# MicroRNAs promote skeletal muscle differentiation of mesodermal iPSC-derived progenitors

Giorgia Giacomazzi[1], Bryan Holvoet[2], Sander Trenson[3], Ellen Caluwé[3], Bojana Kravic[4], Hanne Grosemans[1], Álvaro Cortés-Calabuig[5], Christophe M. Deroose[2], Danny Huylebroeck [6,7], Said Hashemolhosseini[4], Stefan Janssens[3], Elizabeth McNally[8], Mattia Quattrocelli[1,8] & Maurilio Sampaolesi [1,9]

Muscular dystrophies (MDs) are often characterized by impairment of both skeletal and cardiac muscle. Regenerative strategies for both compartments therefore constitute a therapeutic avenue. Mesodermal iPSC-derived progenitors (MiPs) can regenerate both striated muscle types simultaneously in mice. Importantly, MiP myogenic propensity is influenced by somatic lineage retention. However, it is still unknown whether human MiPs have in vivo potential. Furthermore, methods to enhance the intrinsic myogenic properties of MiPs are likely needed, given the scope and need to correct large amounts of muscle in the MDs. Here, we document that human MiPs can successfully engraft into the skeletal muscle and hearts of dystrophic mice. Utilizing non-invasive live imaging and selectively induced apoptosis, we report evidence of striated muscle regeneration in vivo in mice by human MiPs. Finally, combining RNA-seq and miRNA-seq data, we define miRNA cocktails that promote the myogenic potential of human MiPs.

[1] Translational Cardiomyology, Department of Development and Regeneration, KU Leuven, 3000 Leuven, Belgium. [2] Department of Imaging and Pathology, Nuclear Medicine and Molecular Imaging, KU Leuven, 3000 Leuven, Belgium. [3] Department of Cardiovascular Sciences, KU Leuven, 3000 Leuven, Belgium. [4] Institute of Biochemistry, Friedrich-Alexander University of Erlangen-Nürnberg, 91054 Erlangen, Germany. [5] Genomics core, Center for Human Genetics KU Leuven, 3000 Leuven, Belgium. [6] Department of Cell Biology, Erasmus MC, 3015 CN Rotterdam, The Netherlands. [7] Laboratory of Molecular Biology (Celgen), Department of Development and Regeneration, KU Leuven, 3000 Leuven, Belgium. [8] Center for Genetic Medicine, Northwestern University, Chicago, IL 60611, USA. [9] Human Anatomy Unit, Department of Public Health, Experimental and Forensic Medicine, University of Pavia, Pavia 27100, Italy. Mattia Quattrocelli and Maurilio Sampaolesi contributed equally to this work. Correspondence and requests for materials should be addressed to M.S. (email: maurilio.sampaolesi@med.kuleuven.be)

Stem cells hold potential for understanding the regeneration mechanisms with possible applications to degenerative disorders[1]. In particular, the recent advancements in the field of induced pluripotent stem cells (iPSCs) are paving the way to multi-tissue differentiation in patient-matched, isogenic settings[2]. This is particularly compelling for multi-tissue degenerative diseases, such as muscular dystrophies (MDs)[3,4]. MDs encompass a heterogeneous group of inherited myopathies that affect skeletal muscle but in some subgroups also cardiac muscle[5]. At present, no regenerative treatments are available to counteract myofiber and myocyte wastage and functional loss.

The differentiation ability of iPSCs is under the influence of both extrinsic and intrinsic factors[6]. Extrinsic factors that direct differentiation include the addition and withdrawal of specific growth factors. Cell-intrinsic factors include the epigenetic factors that influence the propensity of iPSCs towards the intended lineage[7]. Albeit still unclear, the retention of progeny-specific epigenetic imprinting in iPSCs, e.g., patterns of DNA methylation and histone marks, has been often reported and exploited for enhanced differentiation along the parental lineage[8]. However, it is not yet possible to manipulate the aforementioned epigenetic layers in specific loci. Therefore, the research involving the so-called "epigenetic memory" is still mainly descriptive and the main interventional path resides in the choice of the source cells for iPSC generation[9]. Thus, the search for epigenetic signatures that can be modulated to specifically alter the differentiation propensity of iPSCs, or their derivatives, is still on.

MicroRNAs (miRs) are well positioned to contribute to epigenetic regulation of differentiation of stem cells[10]. The potential of miR-based orchestration of cell fate is evident along the striated muscle lineages and, recently, miRs have been described as part of the epigenetic signature retained after cell reprogramming[11]. MiRs are of particular interest because they and their anti-miRs are small and readily deliverable to manipulate differentiation potential. Better definition of transcriptional and miR profiles will assist in the goal of designing cocktails for skewing the differentiation propensity.

Recently, we described a novel pool of mesodermal, iPSC-derived progenitors (MiPs) for striated muscle regeneration[12,13]. MiPs were sorted as CD140a⁺/CD140b⁺/CD44⁺ cells from differentiating iPSCs of murine, canine, and human origins. Importantly, murine MiPs were able to functionally regenerate both cardiac and skeletal muscles in murine models. Furthermore, the propensity of MiPs toward the skeletal muscle lineage appeared augmented when derived from skeletal muscle mesoangioblasts (MAB-MiPs), as compared to isogenic fibroblast-derived MiPs. The boosted differentiation potential toward skeletal muscle correlated with retained signatures of DNA methylation and histone marks from parental progenies[12]. In this study we investigated the in vivo capacity of human MiPs, mainly focusing our attention on their skeletal myogenic commitment. First, we assessed the translational potential of human MiPs in xenograft-permissive dystrophic mice showing evidence of striated muscle regeneration. Next, we compared the transcriptional profiles of human fibroblast-derived—and MAB-MiPs, and finally we compared their miRNAs profiles in order to predict a miRNA cocktail amenable for modulating the intrinsic propensity and for overcoming the parental lineage retention. We showed that the treatment of MiPs with a selected pro-myogenic miRNA cocktail further improved MiPs contribution to skeletal muscle regeneration.

## Results

### In vivo relevance of human MiPs.
To gain further translational evidence of MiP application, we investigated the regenerative potential of human MiPs in vivo. Importantly, we asked whether the myogenic propensity of human MAB-MiPs, previously shown in vitro[12], was durable in vivo. In addition, we sought to determine whether the human MiPs are necessary for the putative regenerative effect. To address these questions, we equipped previously characterized human, isogenic fibroblast-derived MiPs and MAB-MiPs with GFP, a sodium-iodide symporter (NIS) tracer for non-invasive PET imaging[14], as well as an inducible suicidal gene, iCasp9. This iCasp9 gene triggers apoptosis of cells when exposed to the synthetic inducer AP20187[15]. We first validated the engineered GFP⁺/iCasp9⁺/NIS⁺ (GIN⁺) MiPs in vitro. Suitability for PET imaging was determined by $^{99m}TcO_4^-$ uptake assay (Fig. 1a). Also, cell death was specifically activated within 48 h in GIN⁺ cells only after exposure to the inducer, whereas the inducer alone had negligible effect on control cells lacking iCasp9 (Fig. 1b). We then injected GIN⁺ MiPs in $Rag2^{-/-}$; $\gamma c^{-/-};Sgcb^{-/-}$ mice, which bear skeletal and cardiac muscle degeneration on a xenograft-permissive genetic background[12]. The Sgcb model of limb girdle muscular dystrophy was used because it displays more severe phenotype than the mdx mouse. Each animal received $5 \times 10^5$ cells as an intramyocardial injection in the left ventricle and during the same procedure $5 \times 10^5$ cells in each femoral artery (bilateral). One week post injection, GIN⁺ cells were traceable in the heart and in the hindlimb muscles of cell-treated animals, but not of sham-treated, by PET imaging, although we observed some variability in the detection method. (H and HL fields, Fig. 1c). Semi-quantitative analysis of the standardized uptake value (SUV) showed no engraftment difference between fibroblast-derived MiPs and MAB-MiPs in the heart, whereas hindlimb muscle engraftment was higher (+38.32%) in MAB-MiP-treated animals (Fig. 1d). Four weeks post injection, half of each cohort received intraperitoneal injection of AP20187, while the other half received a vehicle control. The effect of AP20187 was then investigated 8 weeks after the cell injections. Stereo- and immunofluorescence analyses showed that MAB-MiPs engrafted the heart similarly to fibroblast-derived MiPs. Conversely, MAB-MiPs engrafted the hindlimb muscles more efficiently than fibroblast-derived MiPs. Moreover, detection of engrafted fibers was dramatically reduced after AP20187 administration (Fig. 2b). We quantitated MiP-specific contribution to hindlimb stem cells by cytometry-based sorting of the GFP⁺ subfraction of resident CD56⁺ satellite cells and AP⁺ MABs. In both pools, MAB-MiP-treated animals displayed larger GFP⁺ subfractions than fibroblast-MiP-treated ones and GFP⁺ cells were undetectable after AP20187 administration (Fig. 1f).

Differentiation of engrafted MiPs was evaluated using antibodies that detect human but not mouse dystrophin protein (hDYS). hDYS was evident at the sarcolemma following engraftment, and this same pattern was ablated after AP20187 administration (Fig. 2a, b). Four weeks post injection, GFP⁺/hDYS⁺ areas accounted for $34.23 \pm 5.96\%$ and $32.68 \pm 6.83\%$ of the left ventricular wall in fibroblast-derived-MiP and MAB-MiP-injected mice, respectively ($P = 0.62$, $n = 5$, Mann–Whitney U-test). In the gastrocnemius muscles of the same mice, GFP⁺/hDYS⁺ myofibers accounted for $7.25 \pm 0.95\%$ and $22.201 \pm 5.99\%$ respectively ($P < 0.05$, Mann–Whitney U-test). Myofiber measurement at 8 weeks showed that GFP⁺/hDYS⁺ myofibers accounted for $4.23 \pm 1.95\%$ and $14.89 \pm 3.73\%$ in fibroblast-MiP- and MAB-MiP-injected mice respectively ($P < 0.05$ Mann–Whitney U-test).

Regeneration of engrafted striated muscles was quantitated by means of tetanic force measurement, treadmill assay, echocardiography and serum creatine kinase (CK) level monitoring. Four and eight weeks post injection, extensor digitorum longus (EDL) tetanic force and run time values on the treadmill showed that MAB-MiP-treated animals performed better than fibroblast-

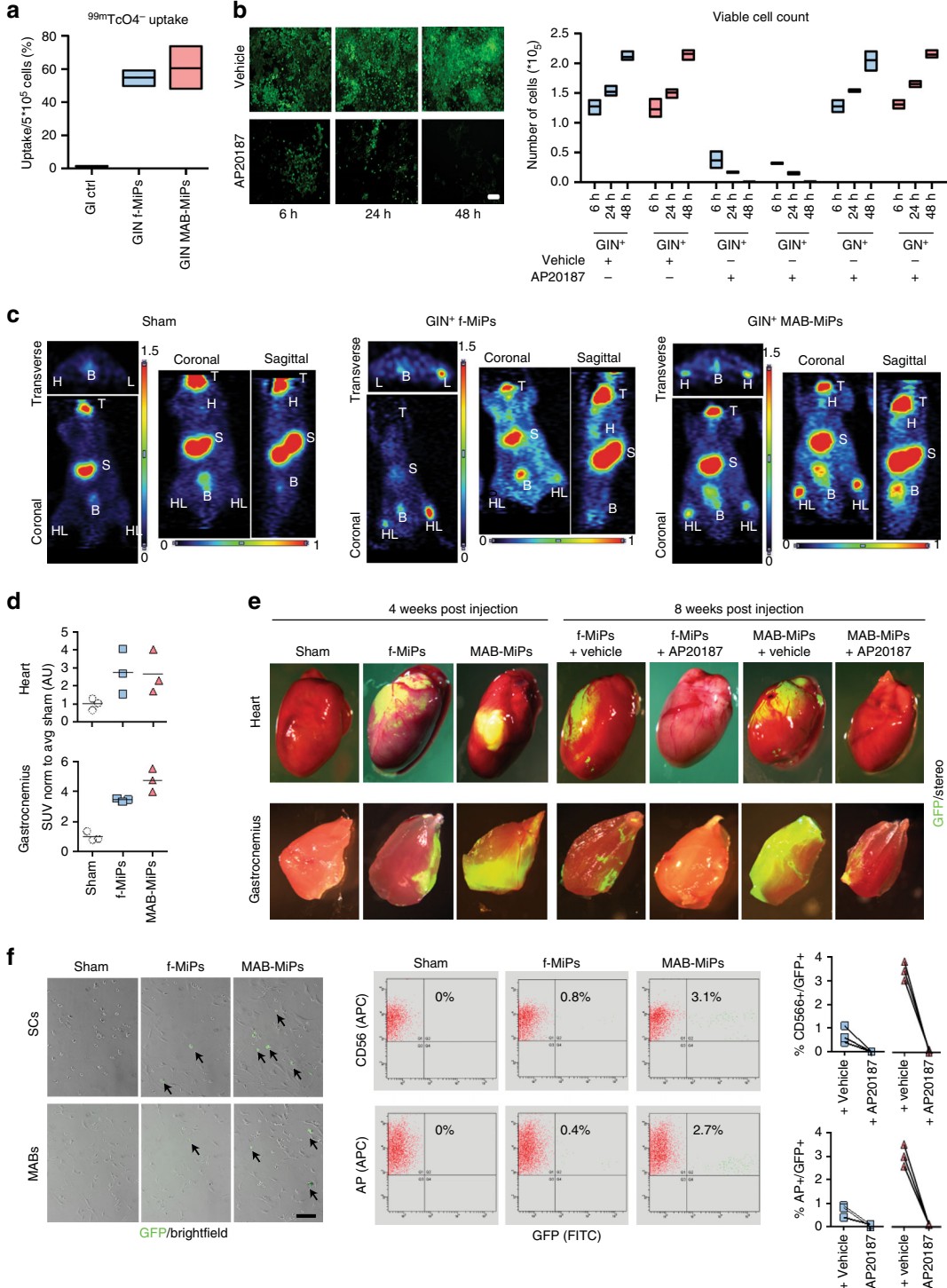

**Fig. 1** Human MiPs display differential myogenic propensity in vivo. **a** Validation of NIS transgene functionality for PET imaging by quantitation of $^{99m}TcO_4^-$ uptake in vitro. *Kruskal–Wallis test and Mann–Whitney $U$-test vs ctrl, $P < 0.05$, $n = 3$/cohort. Scale bar ~100 μm. **b** Validation of iCasp9 transgene by viable cell count after AP20187 administration in vitro. Only iCasp9+MiPs did significantly undergo progressive cell death, which appeared virtually complete after 48 h. *Two-way ANOVA with Bonferroni correction, $P < 0.05$ (interaction), $n = 3$/cohort. Data points depict average and min-to-max span for each sample. **c** 1 week after injection, PET scan of live animals shows engraftment of GIN+MiPs in heart (H) and hindlimbs (HL) muscles of only MiP-treated mice. T, thymus, S, stomach, B, bladder; endogenous positive PET signals. **d** SUV levels in heart and hindlimb regions of PET-scanned mice. *$P < 0.05$ vs sham; **$P < 0.05$ vs sham and fibroblast-MiPs; Kruskal–Wallis and Mann–Whitney $U$ test; $n = 3$/cohort. **e** Stereofluorescence analysis of heart and hindlimb (*gastrocnemius*) muscles of recipient mice before (4 weeks p.i.) and after (8 weeks p.i.) AP20187 administration. **f** Live fluorescence and cytometry analyses of CD56-isolated SCs and AP-isolated MABs from recipient mice at end-point. both SCs and Mabs MAB-MiP-treated animals displayed larger GFP+ subfractions than fibroblast-MiP-treated ones and GFP+ cells were undetectable after AP20187 administration **$P < 0.05$ vs fibroblast-MiP-treated mice; Kruskal–Wallis and Mann–Whitney $U$ test; $n = 3$/cohort at end-point, scale bar ~100 μm

derived-MiP-treated animals (Fig. 2c). Left ventricle fractional shortening was similarly improved by fibroblast-derived MiPs as well as MAB-derived MiPs (Fig. 2d). Serum CK levels were decreased in MiP-treated animals, with lower levels in MAB-MiPs animals (9.7 ± 0.32 U/l; mean ± s.e.m.-) than in fibroblast-derived-MiP-treated (10, 8 ± 0.21 U/l) (Fig. 2d). Notably, all functional parameters were reverted to baseline-like levels after AP20187 administration (Fig. 2c, d). In addition, we evaluated the

morphology of neuromuscular junctions (NMJs) in the skeletal muscles of injected animals. When stained with fluorescent bungarotoxin, NMJs in sham animals appeared fragmented, unlike the normal, contiguous structures found in WT muscles. Fibroblast-MiP- or MAB-MiP-engrafted fibers (discriminated from the non-engrafted according to GFP expression) displayed NMJs with similar morphology to WT and strongly reduced fragmentation (Fig. 2e). Finally we evaluated the extent of fibrotic

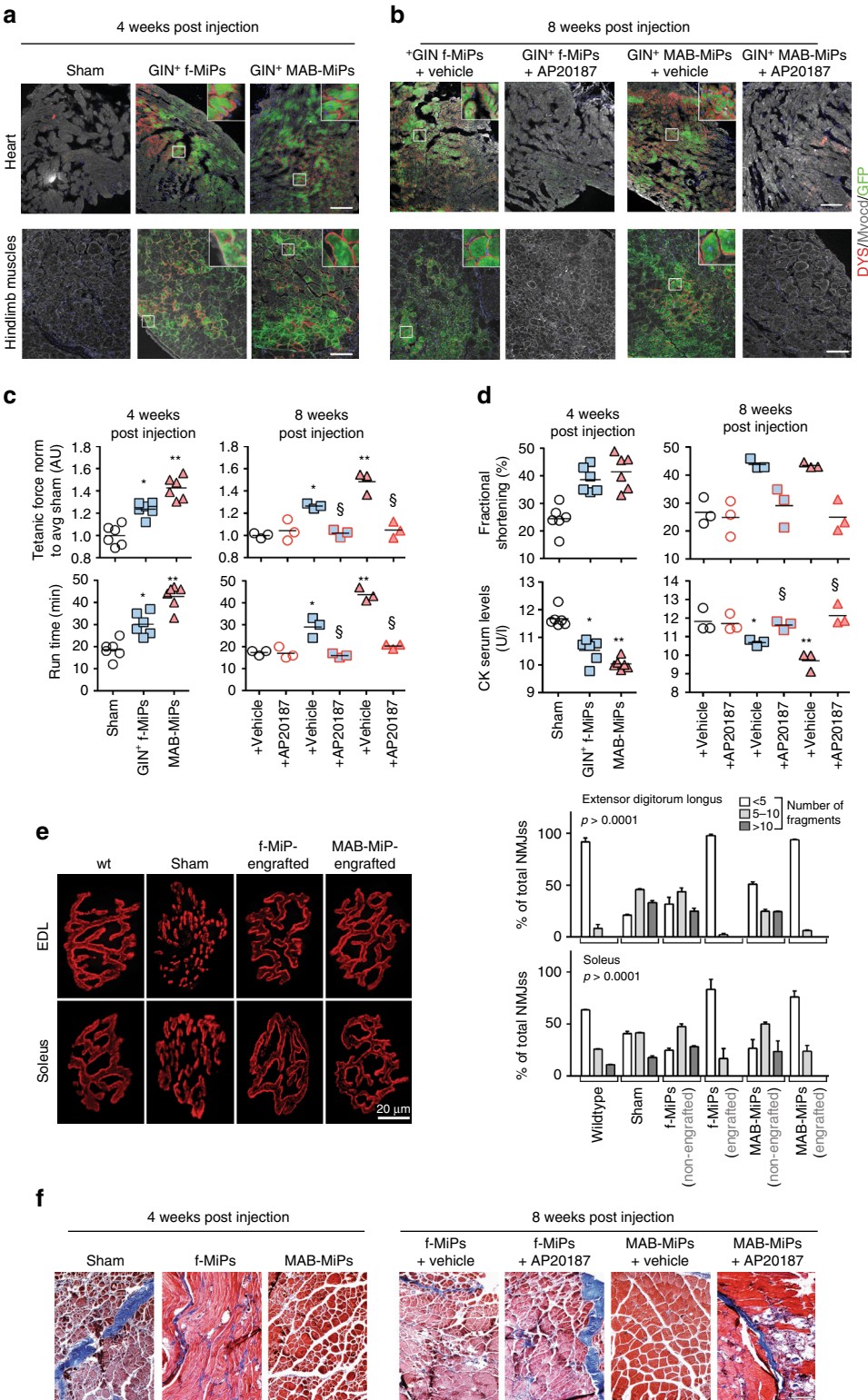

scarring in the skeletal muscle of injected animals. Masson's trichromic staining revealed that fibrosis was reduced to a higher extent in MAB-MiP- than in fibroblast-MiP-treated animals, while it appeared partially reconstituted after AP20187 administration (Fig. 2f). Together, these data suggest that human MiPs have regenerative potential for dystrophic skeletal muscles in vivo. Importantly, the reversal of the beneficial effects after AP20187-induced cell death indicates that MiPs are necessary for exerting the observed regenerative effects. Furthermore, the intrinsic in vivo propensity towards the skeletal muscle lineage was more evident and durable from MAB-MiPs than those derived from fibroblasts.

**Genetic determinants control myogenic potential of MAB-MiPs.** To gain insight in the myogenic difference between MAB- and fibroblast-MiPs, we proceeded to investigate which genes were differentially expressed between fibroblast- and MAB-MiPs, and whether differentially expressed genes were conserved from the parental cells. To this goal, we used RNA-seq to analyze the transcriptional profiles of fibroblast- and MAB-MiPs and the iPSC lines from which the MiPs were derived. For this analysis we used iPSC at the first stage of our differentiation, namely already primed to mesodermal lineages, in order to increase the chance of identifying changes in lineage determination genes. The transcriptional profile of both fibroblast-derived and MAB-derived MiPs was enriched (count > 100 FPKM) in genes associated with myogenic mesoderm formation, including the epigenetic regulators *TET1/2/3, DNMT1/3a, HDAC4, SMARCE1/2/3*, the mesodermal markers *MEF2C, GATA4, TBX3, PAX3, SIX1, ISL1*, and the markers of striated muscle *MYOM2/3, DMD, SGCB*. Conversely, pluripotency-related genes such as *POU5F1, NANOG, ZFP42, DPPA4, LIN28a, GDF3* were poorly detectable or absent (count < 10 CPM; $P < 0.05$ vs all three categories of enriched genes, Wald-log test) (Fig. 3a). Moreover, mRNAs of *AGRIN* and *UTROPHIN* were highly enriched in MiPs (FPKM (avg ± s.d.), 1257.15 ± 191.58 and 1560.39 ± 329.15 in fibroblast- and MAB-MiPs respectively). Unbiased sample clustering and principal component analysis showed that samples clustered primarily according to stage (iPSCs vs MiPs), and secondarily according to progeny (Fig. 3b, c). Intriguingly, gene ontology (GO) comparison of differentially expressed genes at iPSC stage (fibroblast- vs MAB-iPSCs) and MiP stage (fibroblast- vs MAB-MiPs) showed high overlap (79.53%) of Biological Process terms. Overlapping GO terms mainly pertained to developmental program, signaling and cellular metabolism (Fig. 3d and Supplementary Data 1). When clustered on a heatmap, patterns of gene expression emerged reflecting stage-specific (differing iPSCs from MiPs, regardless of progeny) or progeny-specific (differing

fibroblast- vs MAB-derived cells, regardless of stage) (Supplementary Fig. 1a).

We found 905 genes differentially expressed between fibroblast-derived MiPs and MAB-MiPs (Fig. 3e, Supplementary Data 2). Among the significantly differentially expressed genes ($P_{adj} < 0.05$), several myogenic inhibitors were upregulated in fibroblast-derived MiPs, whereas several muscle proteins and myogenesis-associated genes were upregulated in MAB-MiPs. Agonists of BMP signaling such as *BMP6, SMAD5*, and *LTBP4* were upregulated in fibroblast-MiPs, whereas the BMP signaling inhibitor *SMAD7* was upregulated in MAB-MiPs (Fig. 3f). Plotting the fold change of significantly differentially expressed genes in MiPs compared iPSCs, we found that 56.17% of differentially expressed genes conserved the progeny-specific trend across stages, including many identified with the previous analysis (Fig. 3g). We validated a subset of genes important for myogenesis using qPCR. We first compared the expression levels by qPCR between fibroblast- and MAB-progenies at somatic, iPSC and MiP stages. *OSTN, MYB, LHX2, BMP6*, and *SMAD5* were consistently upregulated in fibroblasts, fibroblast-iPSCs and fibroblast-MiPs, while *ANXA3, SMAD7*, and *PAX7* were upregulated in MABs, MAB-iPSCs, and MAB-MiPs. *LTBP4* was upregulated in fibroblasts and fibroblast-MiPs, but not in fibroblast-iPSCs, and similarly *ANXA7* was upregulated in MABs and MAB-MiPs, but not in MAB-iPSCs (Supplementary Fig. 1b).

We then examined CpG methylation and histone mark enrichment in the promoters of these same genes. Bisulfite sequencing analyses revealed that CpG methylation for *OSTN, MYB, LHX2, BMP6*, and *SMAD5* was increased in the MAB progeny, while CpG methylation for *ANXA3, SMAD7*, and *PAX7* was increased in the fibroblast progeny (Fig. 3h). The CpG methylation patterns appeared less discriminative for *LTBP4* and *ANXA7* (Fig. 3h). Chromatin immunoprecipitation analyses revealed that the non-permissive marker H3K9me3 was correlated with the methylation patterns and, conversely, the permissive markers K4me2 and K27ac correlated with the transcriptional upregulation trends (Fig. 3i). Notably, *LTBP4* showed enrichment of K9me3 in the MAB progeny, of K4me2 in fibroblasts and fibroblast-MiPs, and of K27ac in fibroblast-iPSCs, MAB-iPSCs and MAB-MiPs. Conversely, *ANXA7* showed enrichment in K9me3 in the fibroblast progeny and in permissive marks in the MAB progeny, with a spike of K27ac in fibroblast-iPSCs (Fig. 3i). In association with the myogenic propensity shown by MiPs in vivo, we defined *OSTN, MYB, LHX2, LTBP4, BMP6*, and *SMAD5* as an anti-myogenic gene pool, and *ANXA3/7, SMAD7*, and *PAX7* as a pro-myogenic gene pool.

We then asked whether perturbation of these pools could shift the myogenic propensity of fibroblast- and MAB-MiPs. To this end, we combined the endoribonuclease-prepared small

**Fig. 2** Human MiPs engraft and are able to functionally regenerate dystrophic muscles. **a**, **b** Four weeks after delivery, immunostaining shows that fibroblast- and MAB-MiPs engraft and express hDYS to a similar extent in the heart, but differently in the hindlimb muscles (*gastrocnemius*). After AP20187 administration and at the end of treatment (eight weeks post delivery), the trend remains in vehicle-treated mice, whereas GFP and hDYS signals are ablated from AP20187-treated mice. Scale bars, ~100 μm; insets, 20× magnification of indicated field. **c** Tetanic force of EDL muscles and treadmill assay showed that skeletal muscle performance was increased in fibroblast-MiP-treated vs untreated mice, and increased in MAB-MiP- vs fibroblast-MiP-treated mice. Also, the functional gain was lost after AP20187 injection. **d** Functional assessment of the heart, by means of fractional shortening quantitation, shows comparable amelioration in both MiP-treated cohorts, but not after induced MiP death. Furthermore, CK serum levels followed a trend similar to functional assessment of the skeletal muscles. Data points depict the average value of each animal. In **e**, **f**: *$P < 0.05$ vs sham; **$P < 0.05$ vs sham and fibroblast-MiPs; §$P < 0.05$ vs own vehicle-control; Kruskal–Wallis and Mann–Whitney $U$ test; $n = 6$/cohort before AP20187, $n = 3$/cohort at end-point. **e** Bungarotoxin-based staining of NMJs throughout whole EDL and *soleus* muscles ($n = 3$/cohort) reveals that dystrophic mice (sham) present highly fragmented NMJs, while MiP-engrafted fibers present NMJs with a morphology siilar to WT fibers. Quantitation of non-, mildly and highly fragmented NMJs displays that only MiP-engrafted, but not non-engrafted, fibers in treated animals have a WT-like quantitative pattern ($n = 3$/cohort). Depicted are average ± st. dev bars. f-, fibroblast-derived. **f** Masson's trichromic staining of hindlimb (*quadriceps*) muscles of recipient mice. Scale bar ~50 μm. Blue scars denote fibrosis, while myocytes are stained in red. Fibrosis is significantly reduced in MAB-MiPs recepient muscles at 4 weeks p.i., while the difference is ablated after AP20187 amministration at 8 weeks p.i

interfering RNAs (esiRNAs) targeting the anti-myogenic pool in a pro-myogenic cocktail. Conversely, the esiRNAs targeting the pro-myogenic pool were combined in the anti-myogenic cocktail (Supplementary Table 1). Scramble esiRNAs were used as control conditions. Since several key components of the BMP cascade were involved in both gene pools, we enhanced the anti-myogenic

cocktail with soluble BMP6 and the pro-myogenic cocktail with soluble Noggin. We first validated the effects of anti-myogenic and pro-myogenic cocktails on target gene expression. Anti-myogenic pool genes, *LHX2*, *LTBP4*, *MYB*, *OSTN*, and SMAD5, were downregulated in the presence of pro-myogenic cocktail. Similarly, the pro-myogenic pool genes, including *ANXA3*,

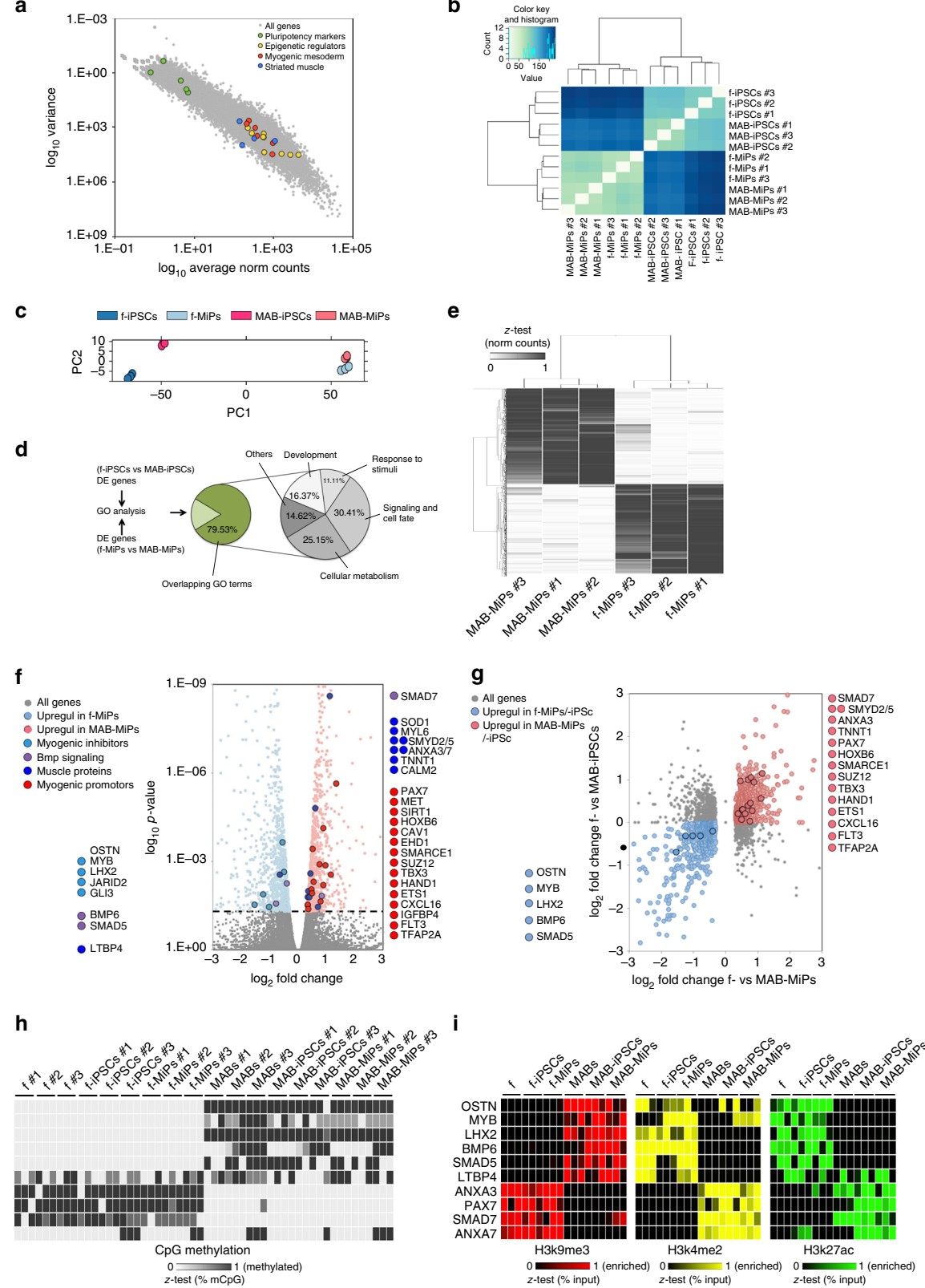

*ANXA7*, *PAX7*, and *SMAD7*, were downregulated in the presence of anti-myogenic cocktail (Supplementary Fig. 1c). We then tested the effect of these cocktails on in vitro myogenic propensity of MiPs in co-culture with C2C12 myoblasts and we proceeded to stain with lamin A/C human nuclei in chimeric myotubes (Supplementary Fig. 1d, e). We observed a significant increase ($P < 0.05$ vs own ctrl (scramble), two-way ANOVA, Kruskal–Wallis test) in the number of myofibers with three human nuclei after PMC treatment compared to cells that received the AMC treatment or to controls. We additionally performed the co-cultures also in presence of the suicidal gene in the MiPs, GIN[+] MiPs (Supplementary Fig. 4a). A higher myogenic propensity would result in higher contribution to chimeric myotubes and hence a higher loss of myotubes after exposure to AP20187. Myogenic propensity was significantly reduced ($P < 0.05$ vs own ctrl (scramble), two-way ANOVA, Kruskal–Wallis test) in fibroblast-derived MiPs and MAB-MiPs after treatment with anti-myogenic cocktail, whereas it was increased in pro-myogenic cocktail-treated cells (Supplementary Fig. 4a). Importantly, pro-myogenic cocktail-treated fibroblast-derived MiPs showed comparable myogenic differentiation to control MAB-MiPs. Thus, perturbation of gene subsets by means of defined factor cocktails enabled perturbation of MiP myogenic propensity. Particularly, reduction of anti-myogenic genes in fibroblast-derived MiPs resulted in a myogenic potential that is comparable to MAB-MiPs.

**Cell type-specific miRNAs regulate myogenic potency of MiPs.** In light of retained miRNA signatures after reprogramming[11], we asked whether miRNAs also influenced the myogenic potential of MiPs. We analyzed the miRNA component of the same samples that we previously analyzed by RNA-seq. The miRNA profile of both fibroblast derived and MAB-MiPs was enriched (count > 30 FPKM) in miRNAs associated with mesodermal progression and myogenesis including *let-7a, miR-1/-590/-497/-34a/-27b/-101/-133a/-138/-15a/-15b/-16/-199a/-21/-22/-221/-23a/-24*. In contrast, miRNAs associated with pluripotency including *miR-372/-302a/-302b/-302c/-302d/-367/106a/-363* were barely detectable or absent in these cells (count < 2 FPKM; Fig. 4a). Similar to the observations in the RNA-seq dataset, unbiased sample clustering and principal component analysis showed stage-specific clustering (iPSCs vs MiPs), with cell type-specific segregation (fibroblast-derived vs- MAB-derived) among each stage (Fig. 4b, c). At the MiP stage, we found 611 differentially expressed miRNAs discriminating fibroblast-derived MiPs vs. MAB-MiPs (Fig. 4d, Supplementary Data 2). Among the significantly upregulated miRNAs in fibroblast-MiPs, we selected those with predicted targets among the upregulated genes previously identified by

RNA-seq. However, we did not experimentally confirm that selected miRNAs are directly targeting the selected differentially expressed genes. Data on the 3′ UTR binding prediction and related mirsvr scores were obtained from microRNA.org (Supplementary Fig. 2). Following this RNA-seq-based filter, we identified *miR-34c-5p/34c-3p/-362/-210/-590* for fibroblast-derived MiPs, and *miR-212/-132/-424/-146b/-181a* for MAB-MiPs (Fig. 4e and Supplementary Fig. 2). Among those, *miR-34c-5p/-34c-3p/-362* and *miR-132/-424/-146b* followed the same differential expression trend seen at the iPSC stage, together with 44.84% of total differentially expressed miRNAs (Fig. 4f). We validated the expression profile of all these miRNAs across somatic, iPSC and MiP stages using qPCR (Supplementary Fig. 3a). As in the previous experiments, we sought to define oligonucleotide cocktails to simultaneously perturb these miRNAs. We combined synthetic inhibitors of fibroblast-MiP-associated miRNAs and synthetic mimics of MAB-MiP-associated miRNAs and tested whether this had pro-myogenic potential. Conversely, inhibitors of MAB-MiP-associated miRNAs and mimics of fibroblast-MiP-associated miRNAs were tested for anti-myogenic potential. These inhibitor and mimics were tested for their effect on target miRNA level expression (Supplementary Fig. 3b). Notably, MiPs treated with this miRNA-based anti-myogenic cocktail showed downregulation of pro-myogenic genes. Those MiPs treated with a miRNA-based pro-myogenic cocktail had decreased levels of anti-myogenic genes (Supplementary Fig. 3c). Consequently, we asked whether the miRNA-targeting cocktails could shift the myogenic propensity of MiPs when co-cultured with C2C12. We found that miRNA-based anti-myogenic cocktail decreased myotube formation while the miRNA-based pro-myogenic cocktail increased the myogenic differentiation. We tested the cocktails in presence and in absence of the apoptotic drug (Fig. 4g, h and Supplementary Fig. 4b). Similarly to what we reported in Supplementary Fig. 1d, e, we observed a significant ($P < 0.05$ vs own ctrl (scramble), Kruskal–Wallis) upregulation of chimeric fibers containing three or more human nuclei in co-cultures that received pro-myogenic treatment compared to cells that received the anti-myogenic treatment or to controls. Finally, we tested the translational relevance of such approach by injecting cocktail-pretreated cells in the femoral artery of $Rag2^{-/-};\gamma c^{-/-};Sgcb^{-/-}$ mice. Four weeks post injection, MiP-specific engraftment and hDYS expression appeared significantly decreased ($P < 0.05$ vs own ctrl (scramble), Kruskal–Wallis) for anti-myogenic-treated cells and increased for pro-myogenic-treated cells, when compared to relative controls (Fig. 4i). Quantitating the regeneration levels by means of hDYS protein levels, we found that pro-myogenic-treated fibroblast-MiPs performed as control MAB-MiPs (Fig. 4j). Thus, we have shown that a defined combination of miRs and soluble ligands

**Fig. 3** RNA-seq shows progeny-specific retention of part of transcriptional profile. **a** Log$_{10}$ plot of variance vs normalized (norm) count for all genes detected in fibroblast- and MAB-MiPs. Data points depict average values of detected genes. Genes associated with pluripotency and with epigenetic/genetic control on muscle mesoderm are highlighted. **b** Unbiased clustering of all MiP and parental iPSC samples analyzed by RNA-seq. **c** PC analysis of all samples reveals stage-specific, progeny-specific clustering. **d** GO analysis (Biological Process) of DE genes between fibroblast- and MAB-derived cells reveals that >79% GO terms are overlapping between MiP and iPSC stages. Sub-categorization of overlapping GO terms is charted at the right. **e** Heatmap (z-test) of DE genes (threshold, >10 norm counts) discriminating fibroblast- and MAB-MiPs. **f** Volcano plot of fold change vs p-value of all DE genes at MiP stage (threshold, $P = 0.05$, dashed line). Left side of the chart, all genes and highlighted candidates enriched in fibroblast-MiPs; right side, genes and candidates enriched in MAB-MiPs. **g** Log$_2$ chart of fold change at MiP stage vs at iPSC stage of significantly DE genes (threshold, $P < 0.05$ at MiP stage). Light blue dots depict genes upregulated in fibroblast-iPSCs and fibroblast-MiPs, while light red dots depict upregulated genes in MAB-iPSCs and MAB-MiPs. Circled dots represent the candidates, as identified in **f**, with conserved DE trend. **h** Quantitation of results obtained from bisulfite sequencing of upstream CpG islands of selected gene shortlists in fibroblast- and MAB-MiPs, and their parental iPSC and somatic cells. Data is depicted as heatmap (z-test) of average percentages of methylated CpGs ($n = 3$ replicates/cell clone). **i** Quantitation of results obtained from ChIP-qPCR experiments analyzing fibroblast- and MAB-MiPs, and their parental iPSC and somatic cells. H3k9me3, repressive mark; H3k4me2, H3k27ac, permissive marks. Data is depicted as heatmap (z-test) of average percentages of IP-enriched vs total input DNA ($n = 3$ replicates/cell clone). f-, fibroblast-derived, DE, differentially expressed

contributes to somatic lineage determination in MiPs. Furthermore, perturbation of defined subsets of miRNAs by means of oligonucleotide cocktails was able to rescue the myogenic potential gap observed in fibroblast-MiPs compared to MAB-MiPs.

Next, in order to unravel additional potential targets of the miRs we performed RNA-seq analysis after the treatment with the selective miRNA cocktails. As shown in Fig. 5a cells clustered according to the treatment that was conducted. We found over 6,000 genes differentially regulated between MAB-MiPs treated

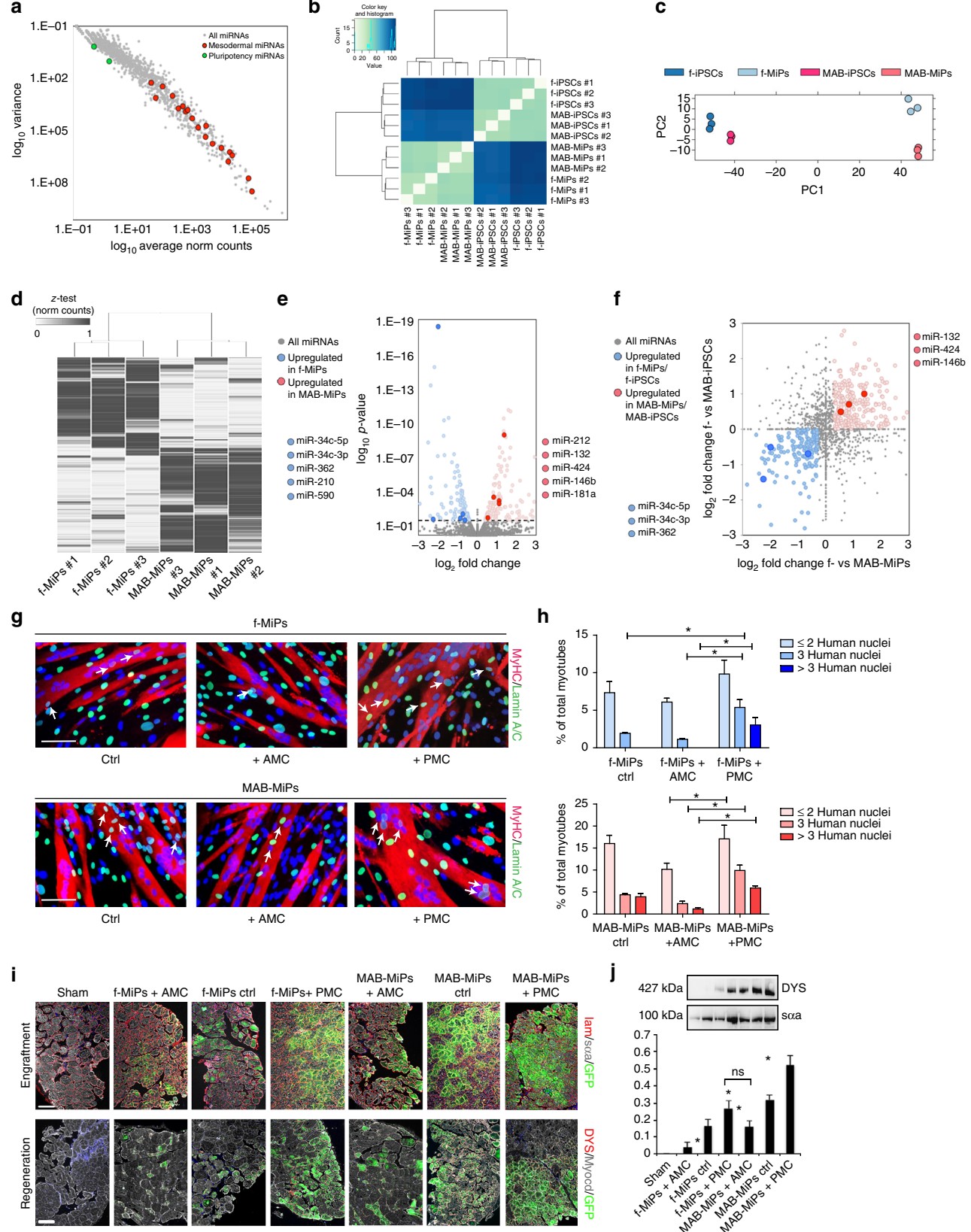

with pro-myogenic cocktails and untreated, and 3,736 genes differentially expressed between MAB-MiPs treated with anti-myogenic vs untreated (Supplementary Data 3). For the fibroblast-MiPs we detected 1,194 differentially expressed genes between pro-myogenic-treated cells and untreated, and 829 between anti-myogenic-treated cells and untreated ($P_{adj} < 0.05$, Wald-log test, Fig. 5, full list in Supplementary Data 3). Consistent with previous analysis ANXA3 was upregulated in MAB-MiPs upon pro-myogenic treatment, while component of the BMP cascade, such as SMAD9 and BMP6 were downregulated in MAB-MiPs treated with pro-myogenic and upregulated in MiPs treated with anti-myogenic cocktails (Fig. 5c–e). LTBP4 was also upregulated in cells treated with anti-myogenic cocktail, whereas MYB was downregulated upon pro-myogenic treatment (Fig. 5d, e). In addition, genes encoding for muscle proteins, including MYL6, ACTA2, TNNT1, were upregulated in pro-myogenic cocktail-treated MiPs, and conversely downregulated in anti-myogenic treated. In these cells myogenic promoters and genes critical for muscle regeneration such as HAND1, CXCL16, MET, and DLL were found downregulated (Fig. 5d). Interestingly, we observed genes involved in smooth muscle differentiation, including CNN2 and CNN3 and other genes relevant for skeletal muscle functionality like MAST1 significantly downregulated in cells that received an anti-myogenic treatment (Fig. 5d, f). Epigenetic regulators, including TET1, CBX6, and HDAC6 were found differentially regulated following the miRNA treatment. Furthermore in cells treated with anti-myogenic cocktails we detected upregulation of several genes that could be involved in the muscle homeostasis, TGFB ligand GDF15 and autophagy related genes ATG10 and ATG14. (Fig. 5e, f).

Finally, we investigated the CpG methylation and the histone markers enrichment in the promoters of the putative pro-myogenic gene pools and anti-myogenic gene pools (Fig. 5g, h). In accordance to the RNA-seq data anti-myogenic genes BMP6, SMAD5, and MYB showed a decrease in methylation in MAB-MiPs treated with anti-myogenic cocktail (Fig. 5g). Furthermore in these cells MYB displayed an increase in permissive histone markers H3k4me2, H3k27ac. When treated with pro-myogenic cocktails, MAB-MiPs showed a decrease of the non-permissive marker H3k9me3 in ANXA3 as well as an enrichment of permissive histone marker H3k4me2 in the promoters of pro-myogenic genes PAX7, ANXA3, and SMAD7 (Fig. 5h).

In fibroblast-MiPs methylation was increased in the promoters of SMAD7 and ANXA7 upon anti-myogenic treatment and this correlates with an increase in H3k9me3 in the promoters of these genes. Conversely anti-myogenic genes such as MYB and BMP6

were found partially methylated and decreased in permissive markers H3k4me2 and H3k27ac after treatment with pro-myogenic cocktail in accordance to what we observed in the RNA-seq data. Taken together our results provide more insight on the anti-myogenic and pro-myogenic miRNA cocktails that we have previously defined, adding additional interesting targets that could be responsible for the differential in vivo performance of MiPs.

## Discussion

Simultaneous regeneration of skeletal and cardiac muscle in dystrophic subjects is compelling, considering that effective repair of skeletal muscle would likely worsen heart conditions[16]. In this regard, MiPs may represent a valid cell tool, as they can be injected in the circulation and efficiently regenerate both striated muscle types[12]. However, two main questions remained unaddressed with respect to the actual translational potential of this iPSC-based strategy: in vivo behavior of human MiPs and whether the myogenic differentiation potential of human MiPs is influenced by reprogrammed cell types or origins, and if external modulators could improve the performance.

To this end we explored the in vivo potential of human MiPs in immunodeficient dystrophic mice in order to determine whether the source of MiPs alters the outcome after engraftment. Taking histological, molecular and functional data together, the capacity to regenerate skeletal muscle of MAB-MiPs appears greater than fibroblast-derived MiPs. Interestingly, the cardiomyogenic potential seems comparable between the two MiP types. These features recapitulate the in vitro behavior previously shown for those cells[12]. In order to document that improvement after engraftment was due to cell-intrinsic effects, we used induced apoptosis in engrafted MiPs and found that this associated with reversal of the beneficial effects at both molecular and functional levels. We cannot exclude that the detrimental effects were partially linked to the drug that induced apoptosis, AP20187. However, this seems unlikely considering that the compound has negligible toxic effects in vivo below 10 mg/kg[17]. Intriguingly, MiP engrafted fibers showed improved NMJ morphology to levels comparable to WT fibers. The mRNAs for AGRIN and UTROPHIN, reported agonists of NMJ formation[18,19], were highly enriched in MiPs. More refined studies are still needed to address the mechanism by which MiPs regenerate the fragmented NMJs in engrafted skeletal muscle fibers.

In this study, we used dual delivery by injecting into both the heart and femoral arteries to remain consistent with previously reported conditions in mice[12]. However, we decided to further

---

**Fig. 4** miRNA-seq analysis allows identification of progeny-specific miRNA cocktails for propensity perturbation. **a** Variance vs norm count plot of all detected miRNAs in fibroblast- and MAB-MiPs. Data points depict average values of detected miRNAs. miRNAs associated with pluripotency and with muscle mesoderm are highlighted. **b** Unbiased clustering of all MiP and parental iPSC samples analyzed by RNA-seq. **c** PC analysis of all samples reveals stage-specific, progeny-specific clustering. **d** Heatmap (z-test) of DE miRNAs (threshold, >10 norm counts) discriminating fibroblast- and MAB-MiPs. **e** Volcano plot of fold change vs p-value of all DE miRNAs at MiP stage (threshold, $P = 0.05$, dashed line). Left side of the chart, all genes and highlighted candidates (miRNAs predicted to bind 3′ UTR of MAB-MiP-upregulated genes) enriched in fibroblast-MiPs; right side, genes and candidates (miRNAs predicted to bind 3′ UTR of fibroblast-MiP-upregulated genes) enriched in MAB-MiPs. **f** $\text{Log}_2$ chart of fold change at MiP stage vs at iPSC stage of significantly DE miRNAs (threshold, $P < 0.05$ at MiP stage). Light blue dots depict miRNAs upregulated in fibroblast-iPSCs and fibroblast-MiPs, while light red dots depict upregulated miRNAs in MAB-iPSCs and MAB-MiPs. Circled dots represent the candidates, as identified in **e**, with conserved DE trend. **g**, **h** Quantification of MiP myogenic propensity in co-culture with C2C12 myoblasts after seven days of differentiation. Myotubes with three or more nuclei were counted as well as human nuclei contributing to chimeric myotubes. Representative fields and quantitation of chimeric myotubes are presented. $P < 0.05$ vs treated and non treated. Kruskal–Wallis and Mann–Whitney $U$ test; $n = 3$ replicates per clone. f-, fibroblast-derived, AMC, anti-myogenic cocktail (gene-based), PMC, pro-myogenic cocktail (gene-based), scale bar ~100 μm. **i** Immunostaining analysis of hindlimb (gastrocnemius) muscles of dystrophic, immunodeficient mice injected with AMC- or PMC-treated MiPs. Upper panels show engraftment, lower panels show appearance of human-specific DYS subsarcolemmal pattern in engrafted fibers. Scale bars, ~100 μm. **j** DYS quantitation by protein analysis is reported on the right. In **g–i**: ns, non significant difference; *$P < 0.05$; Kruskal–Wallis test and Mann–Whitney $U$-test vs own ctrl (scramble oligos), $n = 3$/cohort. Depicted are average ± st. dev bars. f-, fibroblast-derived, DE, differentially expressed, AMC, anti-myogenic cocktail (miRNA-based), PMC, pro-myogenic cocktail (miRNA-based)

explore only the mechanisms to enhance the myogenic potential of MiPs, given the different performance in vitro and in vivo of MiPs towards skeletal muscle, dependent on their cell type of origin, while the cardiac commitment in vivo did not show differences between human f- and MAB-MiPs, consistently with

murine and canine MiPs[20]. Analysis of transcriptional profiles in MiPs confirms that MiPs retain much of the identity of their original cell source. We selected genes to be manipulated to enhance myogenic potential based on their known involvement in MD or in muscle biology. Both *LTBP4*, which was upregulated

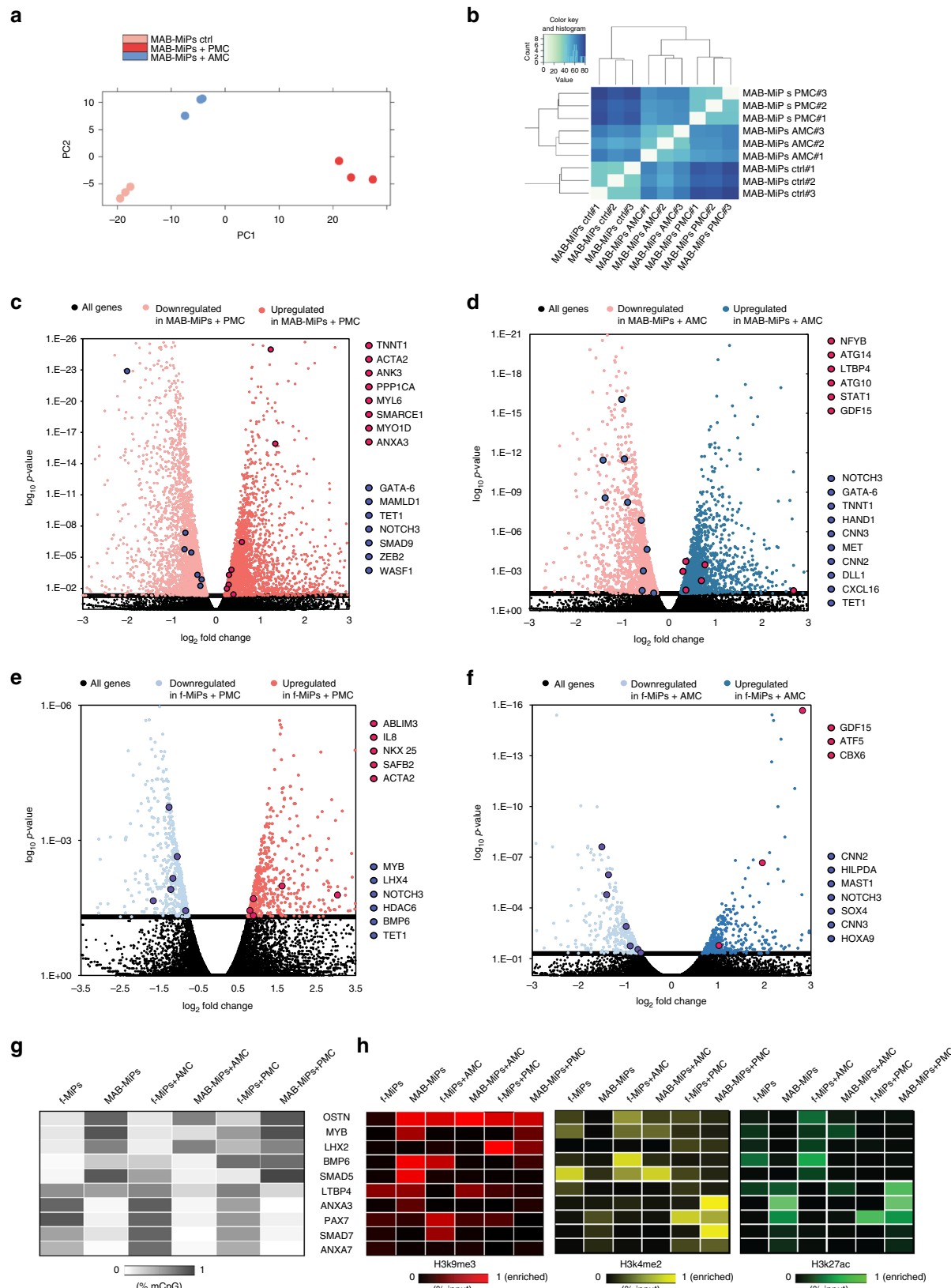

in fibroblast-MiPs, and *ANXA7* which was upregulated in MAB-MiPs, showed progeny-specific differential expression in somatic cells, but not in iPSCs. Considering the epigenetic data, it appears that both genes presented a rather permissive histone signature in our iPSCs. This probably contributed to restoring similar expression levels of *LTBP4* and *ANXA7* in fibroblast- and MAB-iPSCs. Therefore, it may be possible that the progeny-related trends of differential expression for these genes are linked to a non-pluripotent state. *LTBP4* is a TGFβ regulator in the skeletal muscle, and can modify muscular dystrophy[21]. Intriguingly, annexins are also being investigated as genetic modifiers of dystrophic progression[22]. In our experiments, *LTBP4* appeared associated with decreased myogenic potential, while *ANXA7*, together with *ANXA3*, associated with increased propensity. Thus, more focused efforts should be directed on how these factors possibly modify the lineage potential of human MiPs.

Use of miRNAs for enhancing myogenic fate in vivo is particularly compelling and may have high translational potential[23]. To address whether we could identify miRNAs promoting myogenesis, we overlaid RNA-seq and miRNA-seq data with the aim of defining key components that could be used to enhance myogenic potential. In both RNA- and miRNA-seq, the samples cluster primarily according to cell type and this is reflected in the iPSCs and their resulting MiPs. However, MiPs, whether derived from fibroblast or MABs, share many differentially expressed genes but fewer miRNAs. This observation suggests that MiPs may require a more complex body of miRNAs to tightly control a smaller transcriptional divergence. Based on parental cell progeny, we defined combinations of miRNAs to modulate MiP myogenic propensity. Myogenic differentiation in vitro and in vivo was respectively decreased or increased using cocktails of miRs. Notably, a cocktail of pro-myogenic miRs was able to rescue fibroblast-derived MiPs to the degree that they performed similarly to MAB-MiPs. Intriguingly, among the miRNAs we filtered into the anti-myogenic pools, *miR-34c/-362/-210* have been previously associated with pathological state of muscles[24-26] and *miR-590* has been recently associated with differentiation inhibition and the TGFβ pathway[27]. Conversely, within the pro-myogenic pool, *miR-424/-146b/-181a* are associated with myogenesis and muscle development[28-30]. *miR-212/-132* have been shown to inhibit *MECP2*[31], which in turn regulates muscle maturation[32]. The miRNAs tested in the cocktails were selected based on their known functions as well as their differential expression.

RNA-sequencing analysis after anti-myogenic and pro-myogenic miRNA treatment allowed identifying target genes involved in miRNA regulation of MiPs. We found a high number of genes differentially expressed upon miRNA cocktail exposures. Among those, *ANXA3*, *SMAD5*, *BMP6*, *MYB*, *LTBP4* were predicted targets of miRNAs included in the cocktails. In particular exposure to pro-myogenic cocktail determined a consistent downregulation of elements of the BMP/TGFβ signaling pathway, and conversely anti-myogenic exposure resulted in an increased of the TGFβ agonist *GDF15*, that has been interestingly associated with muscle wasting in vivo[33]. Upon pro-myogenic treatment we

detected upregulation of genes coding for proteins important for skeletal (*ACTA2, ABLIM3*) and smooth (*CNN2, CNN3*) muscle homeostasis. These findings suggest potential contributions of the pro-myogenic miRNAs we selected in enhancing smooth muscle differentiation, opening the research of MiPs for appealing new strategies for muscle degenerating diseases, where smooth muscles are highly affected. Consistent with the *modus operandi* of miRNAs many epigenetic modulators were found differentially regulated in treated cells. Interestingly we detected an upregulation of *SAFB2* in pro-myogenic-treated cells. It has been shown that its paralog *SAFB1* facilitates the transition of myogenic gene chromatin from a repressed to an activated state[34]; our results suggest that *SAFB2* could be involved in similar processes. Finally, we have detected upregulation of autophagic markers such as *ATG10* following anti-myogenic treatment, whose aberrant expression has previously been implicated in disease associated with atrophy of the skeletal muscle[35].

Moreover, CpG island methylation and histone markers enrichment analysis after miRNA treatment shed more light on the set of anti-myogenic and pro-myogenic genes that we have put forward. Although CpG island methylation analysis was conducted on the most proximal CpG islands to the transcription start site, this does not indicate that they are directly implicated in gene regulation, nor that they are the only CpG islands responsible for it. Nevertheless, our data suggests that these genes might be epigenetically regulated at the same time by miRNAs, DNA methylation and histone modifications.

In particular anti-myogenic *BMP6*, *SMAD5*, and *MYB* displayed a consistent pattern of DNA methylation and histone markers enrichment, suggesting a simultaneous multifactorial regulation. Other genes, such as *ANXA3* and *LTBP4* did not show univocal signature at epigenetic level upon treatment. Finally, *ANXA7*, *PAX7*, and *SMAD7* promoters' methylation and histone markers enrichment was modulated upon treatment, although we did not detect them among the differentially expressed genes, suggesting once again the multifactorial mechanisms of actions of epigenetic regulators.

We acknowledge that multiple strategies remain to refine and optimize miRNAs useful for promoting myogenesis. In the future, these miRNA-based strategies might benefit muscle regeneration not only based on cell delivery, but also mobilizing and modulating resident stem cells.

This growing knowledge will be fundamental to improve precision and efficiency of miRNA modulation and, ultimately, MiP fate. Moreover, it will be primarily important to test our approach in large animal models, as a proof of a potential scale up. To this end the Golden Retriver Muscular Dystrophy model (GRMD) would provide the ideal fit to investigate cell engraftment, dystrophin restoration and functional rescue of both striated muscle types.

## Methods

**Injection of MiPs and animal models**. Human fibroblast- and MAB-MiPs derived from fetal fibroblasts and MABs[12] were transduced with GFP-iCasp9-[36] and NIS-PuroR-bearing[14] viral vectors. GIN+cells were then sorted for GFP and cultured in

**Fig. 5** RNA-seq after miRNA cocktail exposures shows myogenic-specific committment enhancement of transcriptional profile. **a** PC analysis of a MAB-MiPs samples reveals stage-specific, progeny-specific clustering. **b** Unbiased clustering of MAB-MiPs analyzed by RNA-seq. **c–f** Volcano plots of fold change vs *p*-value of all DE genes in different conditions (threshold, $P = 0.05$, dashed line). MAB-MiPs untreated vs MAB-MiPs+AMC **c**, MAB-MiPs untreated vs MAB-MiPs+PMC **d**, fibroblast-MiPs untreated vs fibroblasts-MiPs+AMC **e**, fibroblast-MiPs untreated vs fibroblasts-MiPs+PMC **f**. Left side of all charts, all genes and highlighted candidates enriched in untreated-MiPs; right side, genes and candidates enriched in treated-MiPs. Blue dots depict genes that were found downregulated while red dots depict genes that were found upregulated upon treatments. **g** Quantitation of results obtained from bisulfite sequencing of upstream CpG islands of selected anti-myogenic and pro-myogenic gene pools in treated and untreated cells. Data is depicted as heatmap of average percentages of methylated CpGs. **h** Quantitation of results obtained from ChIP-qPCR experiments analyzing fibroblast- and MAB-MiPs treated and untreated. H3k9me3, repressive marker; H3k4me2, H3k27ac, permissive markers. Data is depicted as heatmap of average percentages of IP-enriched vs total input DNA ($n = 3$). f-, fibroblast-derived. AMC, anti-myogenic cocktail (miRNA-based), PMC, pro-myogenic cocktail (miRNA-based)

the presence of 1 µg/ml puromycin (Sigma-Aldrich), following reported conditions[12]. All protocols on live mice were performed in compliance with the Belgian law and the Ethical Approval of KU Leuven (P095/2012).

*Rag2-null/γc-null/Sgcb-null* male mice[12] were divided in randomized groups at 3 months of age (*n* = 6, sham; *n* = 6, fibroblast-MiPs; *n* = 6, MAB-MiPs) and injected with GIN+ MiPs (passages 7–10). MiPs were exposed to RPMI20%10% medium for 48 h, then injected in parallel in the left ventricle myocardium and in both femoral arteries under isofluorane anesthesia into each animal ($5 \times 10^5$ cells/ $5 \times 2.5$ µl in the myocardium; $5 \times 10^5$ cells/100 µl per femoral artery). Sham-treated controls received equal treatment and amounts of cell-free saline solution. Engraftment, regeneration and functional outcome were investigated at 4 and 8 weeks post-injection. 1 day after the mid-term analyses, three mice from each cohort were injected with AP20187 ($5 \times 2$ mg/body-kg every other day, i.p.), whereas the other mice of each cohort received vehicle injections.

High-resolution digital ultrasound images were obtained by an experienced echocardiographer using Vevo 2100 Imaging System (Visualsonics) with a 30 MHz probe. Mice were anaesthetized using 1% isofluorane in oxygen, and positioned on the heating pad of the system, in order to maintain normothermia under continuous monitoring. Pre-warmed ultrasound gel was applied on the shaved thorax. B-mode-based 3D reconstruction was carried using the VisualSonics rail system with fixed probe, with ECG- and respiratory gating. FS was calculated based on LVIDd/s values, whereas EDV and CO were calculated based on 3D analysis. Raw data were collected in blind.

Treadmill analysis was conducted on a 10°-uphill oriented treadmill belt with 1 m/min² acceleration on a starting speed of 10 m/min. Mice run was stopped after ≥5 consecutive seconds on the pulsed grill.

Muscle force assessment was performed on freshly isolated EDL muscles upon sacrifice, using a 1200 A in vitro muscle test system (Aurora Scientific). Muscle force was probed upon 20 iterated bouts of isometric contractions (200 Hz, 80 V, 0.5 ms stimulation, 0.5 s tetanus, 10 s interval; 30 °C) in dedicated buffer (1.2 mM $KH_2PO_4$, 0.57 mM $MgSO_4*7H_2O$, 2 mM $CaCl_2*2H_2O$, 10 mM HEPES, 0.5 mM $MgCl_2*6H_2O$, 0.5 mM $MgCl_2*6H_2O$, 4.5 mM KCl, 120 mM NaCl, 0.7 mM $Na_2HPO_4$, 1.5 mM $NaH_2PO_4$, 10 mM D-Glucose, 15 mM $NaHCO_3$; pH 7.3; Sigma-Aldrich). Data were analyzed as % of max absolute force of input sham muscles.

CK levels were measured in resting conditions (>24 h after last treadmill exercise) from serum obtained from >50 µl blood (withdrawn from the tail vein). CK level quantitation was performed using the Creatine Kinase Activity Colorimetric Assay kit (BioVision), following manufacturer's instructions for both sample preparation and standard curve assessment.

**Cell differentiation.** MiP differentiation with C2C12 myoblasts was conducted as previously reported[12]. Briefly cell were seeded 1:10 MiP/myoblast ratio in RPMI 20%/10% medium on collagen-coated vessels for 24 h, then differentiated in DMEM 2% Horse serum medium for 96–120 h in 5% $O_2$/5% $CO_2$ at 37 °C. MiPs and C2C12 were seeded and AP20187 (10 nmol) was added 48 h prior to immunostaining analysis. Myogenic differentiation of human MiPs was conducted seeding 5,000 cells/cm² on gelatin(Millipore)-coated plastic (NUNC) in DMEM-F12 supplemented with 20% FBS and 1% ITS (all reagents from Thermo Fisher Scientific). After 24 h, medium was changed to DMEM-F12 supplemented with 20% FBS, 2 nM SB431542 hyclate and 2 nM LDN193189 hydrochloride (Sigma-Aldrich). After 48 h, medium was changed to DMEM-F12 supplemented with 2% horse serum, 2 nM SB431542 hyclate and 2 nM LDN193189 hydrochloride for additional 48 h, prior to immunostaining analysis.

Gene-targeting cocktails were composed as follows: AMC, esiRNAs anti-ANXA3/-ANXA7/-PAX7/-SMAD7 (Sigma-Aldrich); PMC, esiRNAs anti-OSTN/-MYB/-LHX2/-BMP6/-SMAD5/-LTBP4 (Sigma-Aldrich). Cells were transfected with a total of 1 µg esiRNAs and 1 µl lipofectamine 2000 (Thermo Fisher Scientific) per mw24 well. AMC-treated cells were then kept in medium supplemented with 100 ng/ml BMP6 (Peprotech), whereas PMC-treated cells in medium supplemented with 100 ng/ml Noggin (Thermo Fisher Scientific). Gene expression and differentiation assays were conducted 48 h after transfection. miRNA-targeting cocktails were composed as follows: AMC, miR-mimics for miR-34c-5p/-34c-3p/-362/-210/-590, anti-miRs anti-miR-132/-146b/-424/-212/-181a; PMC, miR-mimics for miR-132/-146b/-424/-212/-181a, anti-miRs anti-miR-34c-5p/-34c-3p/-362/-210/-590 (all oligonucleotides from Sigma-Aldrich). Cells were transfected with 100 nmol of each miR-mimic and 20nmol of each anti-miR per mw12 well, with 2 µl lipofectamine 2000. Gene/miRNA expression and differentiation assays were conducted 48 h after transfection.

**Non-invasive imaging.** Wild type and GIN+ MiPs were plated in quadruplet and incubated with pertechnetate ($^{99m}TcO_4^-$) tracer solution (0.74 MBq/ml in DMEM) for 1 h. Afterwards, cells were rinsed with ice-cold PBS and supernatant was collected. The cells were lysed and collected. The radioactivity of the pellet and supernatant was measured by 2480 Wizard² Automatic Gamma Counter (PerkinElmer, Waltham, MA, USA). The results were adjusted for tracer decay. Uptake values were corrected for cell amounts in the according samples as measured via the NucleoCounter NC-100 system (ChemoMetec, Allerod, Denmark). The mice received an intravenous injection of 3.7–5.55 MBq$^{124}$I (PerkinElmer) on day 3 and an intramuscular injection of 100 µl LASIX (20 mg/ml, Sanofi, Paris, France) as diuretic. At 3 h later, a 20 min static scan was acquired with the Focus 220 small-

animal PET system (Siemens Medical Solutions, Malvern, PA, USA). A transmission scan was acquired using a $^{57}$Co source (185 MBq, Eckert and Ziegler, Berlin, Germany). PET images were reconstructed using a maximum a posteriori (MAP) image reconstruction algorithm and were then analyzed with PMOD 3.0 (PMOD technologies, Zurich, Switzerland) Data were averaged on both legs per animal to keep the intra-injection variability into account. SUV was calculated according to the following formula: SUV=activity concentration in volume of interest/(injected activity/weight of animal). Volumes of interest were manually positioned around the graft regions.

**Molecular and immunostaining assays.** Validation of RNA-seq and miRNA-seq data was carried out by means of qPCR, using SybrGreen for gene levels and Taqman for miRNA levels. Gene qPCR was performed on 1:5 diluted cDNA obtained from 1 µg total RNA (SybrGreen mix, SSIII cDNA production kit and RNA extraction kit from Thermo Fisher Scientific), using Viia7 384-plate reader (Thermo Fisher Scientific; final primer concentration, 100 nM; final volume, 10 µl; *PGK*, internal reference; thermal profile, 95 °C 15 s, 60 °C 60 s, 40×). miRNA qPCR was performed on 1:15 diluted cDNA obtained from 20 ng total miRNA preparation (miRNA isolation kit, Thermo Fisher Scientific). Reagents and probes for reverse transcription and Taqman-based qPCR are from Thermo Fisher Scientific, and manufacturer's protocols were applied.

Methylation patterns were assayed by means of bisulfite sequencing of CpG islands in the promoter regions of target genes (as reported in UCSC genome browser (hg19); primers by MethPrimer). Genomic DNA was isolated through genomic DNA mini kit (Thermo Fisher Scientific), then 1 µg/20 µl was bisulfite-converted using EpiTect Bisulfite kit (Qiagen). Single CpG island amplicons were amplified by PCR (final primer concentration, 330 nM; final volume, 20 µl; thermal profile: 95 °C 30 s, 55 °C 60 s, 72 °C 60 s, 40×, primers are listed in Supplementary Table 2) at T3000 thermocycler (Biometra) using Taq polymerase (Thermo Fisher Scientific), then gel-extracted by means of Gel Extraction kit (Thermo Fisher Scientific) and ligated into pGEM plasmids via TA cloning (Promega). Single bacterial clones were bulk-sequenced (GATC Biotech) and analyzed by means of QUMA online software. Statistical analysis was performed on average methylation values of 5 sequences per cell clone (3 clones per cell type, from independent donors).

Histone mark levels were assayed by means of chromatin immunoprecipitation (ChIP) on the same CpG islands assayed by bisulfite sequencing, adapting previously reported conditions[37] to $5 \times 10^6$ cell pellets. In total, 1 µg/10 µg DNA polyclonal antibodies anti-K9me3 (repressive mark), anti-K4me2 (permissive mark) and anti-K27ac (active mark; all antibodies from Active Motif #39142/ 39133/39765) was used in the ChIP, and protein-A-coated sepharose beads (GE Healthcare) were used for the subsequent pull-down. IgG isotype (eBioscience) was used as negative ChIP control. In total, 5 µl out of 100 µl initial sonicated genomic DNA fragment suspension was used as reference input. Purification of ChIP and input DNA was performed by means of MinElute kit (Qiagen) and quantification as % of input was performed through SybrGreen qPCR, following conditions reported above. Statistical analysis was performed using 3 qPCR replicates per cell clone (3 clones per cell type, from independent donors).

Western blot (WB) analyses were performed on 50 µg cell/tissue lysate (100 µg for DYS analysis) according to commonly used procedures in 10% acrylamide (6% for DYS analysis) hand-cast gels. Here follows the list of antibodies and relative dilutions: mouse anti-sarcomeric α actinin (Abcam #ab72592), 1:500; rabbit anti-GFP (Thermo Fisher Scientific, #A11122), 1:500; mouse anti-DYS3 (interacting with canine and human isoforms, Novocastra #DYS3-CE-S), 1:500. Bands were detected and pictured at Bio-Rad GelDoc by means of Pico substrate (Thermo Fisher Scientific; Dura substrate for DYS analysis). Densitometric analyses were carried on gels loaded, blotted and detected in parallel by means of QuantityOne software (Bio-Rad).

Whole fluorescence imaging of injected tissues was performed at Olympus SZX12 stereomicroscope by means of SISgetIT software (2″ exposition for GFP, 0.2″ for brightfield, semirefringent bottom), and with Zeiss SteREO Discovery V12 microscope by means of AxioImaging software (2″ exposition for GFP, 0.2″ for brightfield, semirefringent bottom). For NMJ imaging and quantitation, mouse soleus and extensor digitorum longus muscles were dissected and fixed in 4% PFA. Muscle fibers were prepared and stained with rhodamine-coupled bungarotoxin using 1:2.500 dilution (rhodamine-BTX, Invitrogen) for 1 h at room temperature. Stained bundles were washed three times and embedded in Mowiol. Z-stacks of individual NMJs were taken with 40× oil objective (Zeiss Examiner Z1). Images were deconvolved and analyzed using 3D deconvolution and 3D measurement modules in AxioVision Software. Data are presented as the mean values, and the error bars indicate ± s.e.m. The number of biological replicates per experimental variable (*n*) is usually *n* > 5 or as indicated in the figure legends. The significance is calculated by unpaired two-tailed *t* test and provided as real *p*-values that are believed to be categorized for different significance levels, like,***$p < 0.001$, **$p < 0.01$, or *$p < 0.05$. Immunofluorescence staining was performed following the commonly used steps of Triton-based (Sigma-Aldrich) permeabilization, donkey serum-(Sigma-Aldrich) background blocking, overnight incubation with primary antibody at 4 °C, 1 h incubation with 1:500 AlexaFluor-conjugated donkey secondary antibodies (Thermo Fisher Scientific), and final counterstain with Hoechst. Here follows the list of primary antibodies and relative dilutions: rabbit

anti-laminin (Sigma #L9393), 1:300; goat anti-GFP (Abcam #ab5450), 1:500; mouse anti-MyHC (DSHB #MF20), 1:3; mouse anti-sarcomeric α actinin (Abcam #ab72592), 1:300; rabbit anti-Myocd (SantaCruz #sc-33766), 1:100; mouse anti-DYS3 (canine- and human-specific, Novocastra #DYS3-CE-S), 1:100; mouse anti-Pax3 (R&D #MAB2457), 1:300; rabbit anti-lamin A/C (#Epitomics 2966-1), 1:600. Imaging was performed at Eclipse Ti microscope (Nikon) by means of Image-Pro Plus 6.0 software (Nikon). Quantitation of engraftment, satellite cell and fiber counts was performed by means of ImageJ software (NIH, USA) on at least 10 fields across heart and muscle samples. SCs and MABs were sorted as $CD56^+$ and $AP^+$ populations at passage 0. Antibodies: mouse anti-CD56 (R&D, FAB7820A), 2 μl/$10^5$cells; mouse anti-AP (R&D, #FAB1448P), 2 μl/$10^5$cells.

**RNA-seq and miRNA-seq.** A $10 \times 10^6$ cell pool per clone was divided in two $5 \times 10^6$ pools for RNA (Total RNA isolation kit and post-isolation DNase, Thermo Fisher Scientific) and miRNA (miRNA isolation kit, Thermo Fisher Scientific) extraction, respectively. RNA (>10 μg) and miRNA (~1 μg) samples were then verified and processed by the Genomics Core (KU Leuven – UZ Leuven, Belgium). RNA-sequencing libraries were constructed with the TruSeq RNA Sample Prep Kits v2 (Illumina). RNA-sequencing after miRNA treatment was performed 48 h after treatment (n = 3 per conditions). Three RNA samples per group (one per clone) were indexed with unique adapters and pooled for single read (50 bp) sequencing in Illumina HiSeq2000. RNA-seq reads were aligned with TopHat v2.0.2 to the human genome version hg19[38]. Transcripts were assessed and quantities were determined by Cufflinks[38]. Differential expression levels were assessed using DESeq[39] and Wald-log test was applied. GO analysis was performed by means of BinGO (biological process; within Cytoscape 3.2.1) on DE genes in fibroblast- vs MAB-iPSCs, and in fibroblast- vs MAB-MiPs comparisons. GO terms with $P < 0.05$ were subsequently compared between iPSC and MiP stages. Variance, count and fold change analyses, as well as charts and z.test matrices were conducted by means of Excel software (Microsoft). Heat maps were obtained analyzing the z.test value matrices with GITools 2.2.2[40] (hierarchical clustering, Manhattan distance, average linkage). Data about 3′UTR binding prediction and related mirsvr scores were obtained from microRNA.org[41,42] (Aug 2010 release; conserved, good-score matrix).

**Statistical analyses.** Sample size for in vitro/in vivo experiments was calculated by means of Sample Size Calculator (http://www.stat.ubc.ca/~rollin/stats/ssize/index. html; parameters: power, 0.80; alpha, 0.05). When applicable, sample size analysis was based on average values obtained from preliminary optimization/validation trials. When comparing multiple data pools, Kruskal–Wallis test followed by Mann–Whitney U test between two target populations were applied and significance was scored when $P < 0.05$ for both tests. When comparing two data pools, Mann–Whitney U test was applied and significance scored when $P < 0.05$. All statistical analyses were conducted using Prism v5.0 (GraphPad).

**Study approval.** All protocols and experiments on mice and murine samples were performed in compliance with the Belgian law and the Ethical Approval of KU Leuven.

**Data availability.** The authors declare that all data supporting the findings of this study are available within the article and its supplementary information files or from the corresponding author upon request. RNA-seq and miRNA-seq data have been deposited in GEO NCBI under the accession code GSE102283.

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

## Acknowledgements

This work has been supported by FWO (#G060612N, #G0A8813N, #G088715N), Opening the Future Campaign EJJ-OPTFUT-02010, Excellentiefinanciering KUL Project grant and Rondoufonds voor Duchenne Onderzoek to M.S., to E.M. and AFM#18373, FWO#1263314 to M.Q. S.H. receives support by a grant from the German Research Foundation (HA 3309/1-3) and the Interdisciplinary Center for Clinical Research Erlangen (E17). M.S. and D.H. are supported by KU Leuven Research Council funding (OT-09/053) and GOA-11/012), the Belgian Agency for Science Policy (Belspo) network IUAPVII-07 DevRepair. G.G. and M.Q. are grateful to Andy Vo for critically reviewing the ms, and to Ruben Dries for valuable suggestions about RNA-seq data analysis.

## Author contributions

M.Q., G.G. and M.S. designed research and drafted the ms; G.G., M.Q., B.H., S.T., E.C., B.K. and H.G. performed experiments; G.G., M.Q., C.D., D.H., S.H., S.J., A.C.C. and E.M. analyzed data, contributed to critical experiments/insights; M.Q., E.M. and M.S. provided funding.

## Additional information

**Competing interests:** The authors declare no competing financial interests.

