## [Peer Review File · Nature Communications]

Reviewers' comments:

Reviewer #1 (Remarks to the Author):

The manuscript by Giacomazzi et al describes the cardiogenic and myogenic potential of human mesodermal iPSC derived progenitors (MiPs) and compare these potential between iPSC derived from mesangioblast and fibroblast. This is the follow-up of their report in JCI in 2015, where human MiPs were only tested in vitro. In a second part, they investigate sets of gene and of miRNAs that could have a pro-myogenic or anti-myogenic effect. The second part is highly original and should be of importance in the general field of iPSC differentiation into skeletal muscle, which has been problematic in the last years, and most of the reports published were not followed by the groups or other groups, probably due to a lack of reproducibility. This looks much more promising than what I have seen up to now.

The manuscript is definitely interesting, the results in general are rather convincing, and it could be of general enough interest for readers of Nature Com. However, as a general comment, the common rationale between the two parts is not very well presented and could be improved to better articulate these two parts. The first deals with both myogenic and cardiogenic potential of the MiPs, while the second part is focused only on the myogenic potential. The authors should explain better what is the general aim of the manuscript.

A second general comment is that the pdf version of the figures that was transferred to the reviewers has several problems. As an example, on Figure 1 there are 2 pictures overlaid on panel E that in fact come from figure 2 ! Another example is the absence of supplementary Figure 2C and D in the file provided to the reviewer, as well as of supplementary Figure 4, while they are described in the result section. This may result from a problem while generating the pdf, and not due to the authors themselves, but it was difficult to assess some parts of this manuscript in these conditions.

Specific comments:

- Figure 1B: The quantification is very hard to understand, since it is minimally described in both result and figure legend sections. Why using a ratio to 0 hrs and what does it mean? Why not simply use the number of cells?
- Figure 1C: The authors describe a bilateral injection in the femoral arteries. However, on the version of the figure the reviewer had, there seem to be a much stronger signal on one side as compared to the other. Is this representative? Could the authors comment on this?
- Figure 1D: The plots show only 3 points per condition, while the material and method section indicates that 6 animals were used per condition.
- Regarding the effect of AP20187, the text refers to Figure 1E and 2A, but AP20187 appears in figure 2B, and this is at 8 weeks while the text mentions only 4 weeks time point.
- Figure 2 A and B: The dystrophin signal in skeletal muscle seems to be weaker and less dispersed at 8 weeks compared to 4 weeks. Is that just an impression? Or could it correspond to a loss of fibres containing human nuclei between these two time points? Could the authors comment on this? The % of fibres is only calculated at 4 weeks. Is it possible to get it at 8 weeks? And why it has been done only on gastrocnemius and not on other hindlimb muscles?
- There is an error in the figure legend of Figure 2: panel C appears twice and D should be E.
- CK analysis: the serum levels are clearly decreased, but it is hard to see on the figure of how much. Could the authors provide in the text mean values and standard deviations for these CK levels?
- End of page 9: The authors enhanced their anti and pro-myogenic cocktails with BMP6 and Noggin respectively. What is the rationale for doing so? What is the question addressed? And how to sort out the effect of the cocktail from the effect of BMP6 or Noggin?
- Page 10: in vitro experiments. The authors use the loss of myotubes after treatment with AP20187 as readout, but it would have been more straightforward (and probably more accurate) to use a marker of human nuclei. There are several antibodies specific for human nuclei on the market, including a human specific anti lamin A/C widely used in xenotransplant experiments.
- End of page 11: The authors show that treating cells with their cocktail of miRs can modulate the fate of iPSC, but the conclusion stating that miR participate in the somatic lineage determination in

MiPs seems a bit overstated. miR can modulate the somatic lineage determination when applied on these cells, but that does not prove that they do it without external intervention ! The same is true for the title of the manuscript: microRNA CAN modulate myogenic propensity would be more in accordance with the results described here.

- There is a problem of edition in the end of the first paragraph of the discussion, which mentions 3 questions, but only 2 are specified, and the second question (whether the regenerative functional effects are strictly dependent either on the capacity of MiP's to differentiate) is impossible to understand.

Reviewer #2 (Remarks to the Author):

This is a follow-up study of an article published by the same group in 2015 in the Journal of Clinical Investigation. In the JCI paper, they had described a novel pool of mesodermal iPSC-derived progenitors (MiPs). When isolated from skeletal mesangioblasts (from either murine or canine origin), and after engraftment into dystrophic immunodeficient mice, these cells are able to differentiate into both cardiac and skeletal muscle. However, when isolated from fibroblasts, their differentiation potential into skeletal muscle is strongly decreased.

In this new article, the authors have performed the same experiments, but MiPs cells are from human origin. Fibroblast- or MAB-MiPs were grafted into immunodeficient dystrophic mice and again, the MAB-MiPs engrafted the skeletal muscles more efficiently than the fibroblast-derived MiPs. An improvement in muscle strength and muscle function is also observed.

The authors have also identified genes differentially expressed between F-MiPs and MAB-MiPs. Among these genes, myogenic inhibitors were up-regulated in F-MiPs, whereas myogenic-associated genes were up-regulated in MAB-MiPs. These different transcriptional profiles are due to CpG methylation differences and histone mark modifications in the promoters of these genes, and different miRNA expressions targeting these genes.

The manuscript submitted for review is well written but, it would need further experiments before it can be considered for publication.

The authors have identified 905 genes differentially expressed between f-MiPs and MAB-MiPs but the entire list is not provided. Several genes have been chosen for further experiments, but why these genes in particular? How were they selected? Is it possible to analyze the epigenetic modifications within the entire genome to more precisely determine the epigenetic alterations that would be important for efficient muscle engraftment?

Concerning the CpG experiments, the primer used are not indicated. In the materials and methods, the authors indicate the primers were chosen in the promoter regions reported in the UCSI genome browser (UCSC I presume?). How many CpG were analyzed? Are the promoters well characterized? Do they know if the CpGs they have looked at are implicated in gene regulation?

None of the Suppl Figures is well numbered. In Suppl 2 (which is actually Suppl Figure 1 according to the figures), the sequences of the esiRNAs are not indicated. Can the authors clarify the sentence p9: "we combined the esiRNAs targeting the anti-myogenic pool in a pro-myogenic cocktail". How did they choose BMP6 and Noggin?

In Suppl 2 D and Fig 4G, the experimental protocol is complicated. Co-cultures were realized between C2C12 and the MiPs carrying a suicidal gene. Myogenic potentials of the MiPs were assessed by quantification of MyHC-positive area after exposure to AP20187 which induces death of MiP-C2C12 myotubes. There are 2 important biases in the analysis: 1) the test does not make the difference between a myotube exclusively composed of human nuclei and myotubes composed of both human and murine nuclei. 2) Differentiated mononuclear cells can be stained by the MF20 antibody.

It would have been more appropriate to co-culture the cells and performed the MF20 staining without AP20187 exposure. The number of myotubes containing 3 or more nuclei would have been

counted as well as the number of human nuclei per myotube. Human and murine nuclei can be very easily differentiated: C2C12 nuclei are characterized by the presence of heterochromatin revealed after Hoechst treatment or/and human nuclei are specifically recognized by the human specific LAMIN A/C antibody.

MF20 labelling is not clear enough in Suppl 2. In Fig 4, in the presence of the vehicle, why is there no difference between f- and MAB-MiPs in the control wells? According to the previous article, human MAB-MiPs should show greater myogenic potential in co-culture with C2C12 than f-MiPs. This does not seem to be the case here and the way the graph is presented does not allow the reader to appreciate the intrinsic differences between the different points. In both Suppl 2 and Fig 4, in the MyHC+ graph, I would recommend to not normalize each control at 1 in order to be able to compare all the samples and to perform the adequate statistical test.

In the miRNA part, the authors have found 611 differentially expressed miRNAs but again, the whole list is not provided. They have selected the miRNAs with predicted targets among the up-regulated genes previously identified per RNA-seq but the experiments performed to demonstrate the miRNAs really target these genes and are able to modulate their expression are not clear. Was it at mRNA level? Protein level? Both? Moreover, since one miRNA is able to modulate the expression of several genes, it would have been interesting to perform a RNA-seq on the treated cells to more precisely determine the genes that would be important for efficient muscle engraftment. A CpG/histone modification analysis after MiPs treatment with the miR and anti-miR cocktail might be interesting.

In the discussion, the authors should discuss the fact that the genes they have selected may be regulated by DNA methylation, histone modification and microRNA at the same time.

There are a huge numbers of errors in the manuscripts.

In the abstract: "we document that wildtype and genetically corrected MiPs can successfully engraft into the skeletal muscle and hearts of dystrophic dogs. ». There is nothing like that in the article. No dogs, no genetically corrected cells!!

The Supplementary figures are not well numbered. Moreover, the Suppl 2 is not mentioned in the text.

In figure 4, there is no point J. In Fig. 1, there is an unwanted picture right in the middle of the figure.

In figures, 1B, 1G and Suppl 1D, the text and the lines are not well aligned with the graphs.

In the material and methods, in the study approval section, "Injection of GRMD dogs and collection of canine samples was performed in compliance with the French law and the Ethical Approval of Maison Alford Veterinary School." Again, there is no dog in this article.

Itemized reply

We are grateful for the reviewers' comments that greatly contributed to improve the quality of our paper in its current form. Answers to the referees are reported in *red italics*.

Reviewers' comments:

Reviewer #1 (Remarks to the Author):

The manuscript by Giacomazzi et al describes the cardiogenic and myogenic potential of human mesodermal iPSC derived progenitors (MiPs) and compare these potential between iPSC derived from mesangioblast and fibroblast. This is the follow-up of their report in JCI in 2015, where human MiPs were only tested in vitro. In a second part, they investigate sets of gene and of miRNAs that could have a pro-myogenic or anti-myogenic effect. The second part is highly original and should be of importance in the general field of iPSC differentiation into skeletal muscle, which has been problematic in the last years, and most of the reports published were not followed by the groups or other groups, probably due to a lack of reproducibility. This looks much more promising than what I have seen up to now.

The manuscript is definitely interesting, the results in general are rather convincing, and it could be of general enough interest for readers of Nature Com. However, as a general comment, the common rationale between the two parts is not very well presented and could be improved to better articulate these two parts. The first deals with both myogenic and cardiogenic potential of the MiPs, while the second part is focused only on the myogenic potential. The authors should explain better what the general aim of the manuscript is.

We thank the reviewer for appreciating our work, for these comments and observations. We have edited the ms to better highlight the general aims of this study. Specifically, see introduction (line 72-75 page 5) and discussion (333-337 page 15).

A second general comment is that the pdf version of the figures that was transferred to the reviewers has several problems. As an example, on Figure 1 there are 2 pictures overlaid on panel E that in fact come from figure 2 ! Another example is the absence of supplementary Figure 2C and D in the file provided to the reviewer, as well as of supplementary Figure 4, while they are described in the result section. This may result from a problem while generating the pdf, and not due to the authors themselves, but it was difficult to assess some parts of this manuscript in these conditions.

We apologize for the errors while generating the pdf that was given to the reviewers. We have now edited the ms and corrected the figures.

Specific comments:

- Figure 1B: The quantification is very hard to understand, since it is minimally described in both result and figure legend sections. Why using a ratio to 0 hrs and what does it mean? Why not simply use the number of cells?

We apologize for lack of clarity and we revised Figure 1B. Actually the normalization was related to the number of cells at the beginning of the experiments (0h) that was 100.000 cells for each sample. Thus the y-axis can refer to $\times 10^5$ cells. We changed the axis label in the revised Figure 1B. Anyway after administration of the vehicle the number of cells increases after 6, 24, 48 hours, while upon administration of the drug the ratio drops at all data points.

- Figure 1C: The authors describe a bilateral injection in the femoral arteries. However, on the version of the figure the reviewer had, there seem to be a much stronger signal on one side as compared to the other. Is this representative? Could the authors comment on this?

This is due to the variability of the injection itself, although performed always by the same researcher performance can vary. The purpose of this panel is to show that there is indeed signal after one week at the hindlimbs and at the heart, however for more accurate analysis please refer to the further analysis described in the manuscript. A comment on variability of this detection method, probably border line with the minimal amount of cells for the signal is added in the results section (line 102 page 6).

- Figure 1D: The plots show only 3 points per condition, while the material and method section indicates that 6 animals were used per condition.

6 animals per condition were injected in total to be used for all the studies. However, only 3 animals in each cohort were scanned (as reported in the legend of Figure 1D, n=3 in each cohort, line 718-719, page 30). This is due to technical reasons, since the animals used are immunodeficient and dystrophic and therefore very sensitive to repeated general anesthesia.

- Regarding the effect of AP20187, the text refers to Figure 1E and 2A, but AP20187 appears in figure 2B, and this is at 8 weeks while the text mentions only 4 weeks time point.

We thank the reviewer for picking up our mistake in referring to the figures in the text. The effect of the administration of the drug AP20187 can be appreciated indeed in Figure 2B on hindlimb muscles. The drug was indeed administered at 4 weeks post injections, however the effect was investigated after 8 weeks from the injections, so 4 weeks from the administration of the drug as now reported in line 105-108, page 6.

- Figure 2 A and B: The dystrophin signal in skeletal muscle seems to be weaker and less dispersed at 8 weeks compared to 4 weeks. Is that just an impression? Or could it correspond to a loss of fibres containing human nuclei between these two time points? Could the authors comment on this? The % of fibres is only calculated at 4 weeks. Is it possible to get it at 8 weeks? And why it has been done only on gastrocnemius and not on other hindlimb muscles?

The signal of dystrophin in skeletal muscle is indeed decreased at 8 weeks p.i (line 121-122, page 7). This can be due to a loss of fibers between the two time points, due to the fact that the amount of donor dystrophin is not comparable to the wt resulting in weaker muscle fibers.

However, there is still a considerable number of surviving myofibers that can be appreciated from figure 2B. According to previous studies (Sampaolesi et al 2003) the gastrocnemius represents a good indicator of cell engraftment (2.1% of cell engraftment compared to 3.1% and 0.8% of quadriceps and tibialis anterior respectively).

- There is an error in the figure legend of Figure 2: panel C appears twice and D should be E.

Once again we apologize for this error that has now been corrected.

- CK analysis: the serum levels are clearly decreased, but it is hard to see on the figure of how much. Could the authors provide in the text mean values and standard deviations for these CK levels?

We have now provided in the text (line 129-130, page 7) mean values and s.e.m. for CK levels, as follows: MAB-MiPs animals (9.7 ± 0.32 , U/l \pm s.e.m.), fibroblast-MiPs animals ($10,8 \pm 0.21$)

- End of page 9: The authors enhanced their anti and pro-myogenic cocktails with BMP6 and Noggin respectively. What is the rationale for doing so? What is the question addressed? And how to sort out the effect of the cocktail from the effect of BMP6 or Noggin?

BMP6 and SMAD5 (BMP agonists) were upregulated in the f- progeny; conversely SMAD7 (BMP antagonist) was downregulated in the MAB progeny. Instead of upregulating the pathway with ligand gene expression, soluble BMP6 agonist was used in the antimyogenic cocktail. Correspondingly, a protein inhibitor of the pathway (Noggin) was used in the promyogenic cocktail. In combination with the RNAseq data, agonism and antagonism of the pathway with defined protein factors reliably implicates the BMP pathway in MiP regulation, although within the multi-factorial context of compound cocktails.

- Page 10: in vitro experiments. The authors use the loss of myotubes after treatment with AP20187 as readout, but it would have been more straightforward (and probably more accurate) to use a marker of human nuclei. There are several antibodies specific for human nuclei on the market, including a human specific anti lamin A/C widely used in xenotransplant experiments.

According to the reviewer 1 and 2 suggestions we performed again the analysis including the number of myotubes (>3 nuclei) in each condition and the number of human nuclei per myotube. In this analysis we excluded MF20+ mononucleated cells as reported below and as Figure 4 and Suppl Figure 1 in

the revised ms.

- End of page 11: The authors show that treating cells with their cocktail of miRs can modulate the fate of iPS, but the conclusion stating that miR participate in the somatic lineage determination in MiPs seems a bit overstated. miR can modulate the somatic lineage determination when applied on these cells, but that does not prove that they do it without external intervention ! The same is true for the title of the manuscript: microRNA CAN modulate

myogenic propensity would be more in accordance with the results described here.

We thank the reviewer for this remark. We have indeed revised the manuscript and the title in light of this comment (line 260-262, page 12).

- There is a problem of edition in the end of the first paragraph of the discussion, which mentions 3 questions, but only 2 are specified, and the second question (whether the regenerative functional effects are strictly dependent either on the capacity of MiP's to differentiate) is impossible to understand.

We apologize for not having edited this part before submission. Indeed there are only two questions that should be mentioned. We have now revised this part of the manuscript in what we hope is a more straightforward message (line 314-317, page 15).

Reviewer #2 (Remarks to the Author):

This is a follow-up study of an article published by the same group in 2015 in the Journal of Clinical Investigation. In the JCI paper, they had described a novel pool of mesodermal iPSC-derived progenitors (MiPs). When isolated from skeletal mesangioblasts (from either murine or canine origin), and after engraftment into dystrophic immunodeficient mice, these cells are able to differentiate into both cardiac and skeletal muscle. However, when isolated from fibroblasts, their differentiation potential into skeletal muscle is strongly decreased.

In this new article, the authors have performed the same experiments, but MiPs cells are from human origin. Fibroblast- or MAB-MiPs were grafted into immunodeficient dystrophic mice and again, the MAB-MiPs engrafted the skeletal muscles more efficiently than the fibroblast-derived MiPs. An improvement in muscle strength and muscle function is also observed.

The authors have also identified genes differentially expressed between F-MiPs and MAB-MiPs. Among these genes, myogenic inhibitors were up-regulated in F-MiPs, whereas myogenic-associated genes were up-regulated in

MAB-MiPs. These different transcriptional profiles are due to CpG methylation differences and histone mark modifications in the promoters of these genes, and different miRNA expressions targeting these genes.

The manuscript submitted for review is well written but, it would need further experiments before it can be considered for publication.

The authors have identified 905 genes differentially expressed between f-MiPs and MAB-MiPs but the entire list is not provided.

We thank the reviewer for these comments and observations and we now included the list of the genes as Supplementary Table 1 in the revised ms.

Several genes have been chosen for further experiments, but why these genes in particular? How were they selected? Is it possible to analyze the epigenetic modifications within the entire genome to more precisely determine the epigenetic alterations that would be important for efficient muscle engraftment?

We have chosen the selected genes via data mining of mRNA-seq and the miRNA-seq combined, with the rational of looking at possible match and interactions. Epigenetic modifications within the entire genome are still very costly and beyond the scope of our study and will require more than 6months. However we followed the suggestion of the reviewer to perform additional genome wide analysis and we carried out a second RNA sequencing on the miRNA treated samples as requested (see below) and we verified the epigenetic modifications in target genes (CpG island methylation and histone mark analyses). Acknowledging that epigenetic modifications are multifactorial processes, it was our intent to primarily concentrate our investigation on one of the regulators that might play a role in MiPs-driven muscle regeneration, namely microRNAs.

Concerning the CpG experiments, the primer used are not indicated. In the materials and methods, the authors indicate the primers were chosen in the promoter regions reported in the UCSI genome browser (UCSC I presume?). How many CpG were analyzed? Are the promoters well characterized? Do they know if the CpGs they have looked at are implicated in gene regulation?

We have now included a list of primers used for CpG island analysis in Supplementary table 1. We indeed used UCSC genome browser to select the regions for our studies. We have designed primers via MethPrimers that encompass at least 8 and up to 18 CpG per promoters. We have chosen the most proximal CpG region to the TSS, relying on the information provided by UCSC genome browser on what is already known about these promoters.

None of the Suppl Figures is well numbered. In Suppl 2 (which is actually Suppl Figure 1 according to the figures), the sequences of the esiRNAs are not indicated. Can the authors clarify the sentence p9: “we combined the esiRNAs targeting the anti-myogenic pool in a pro-myogenic cocktail”. How did they choose BMP6 and Noggin?

We thank the reviewer for this remark, we have now included the target sequences of the esiRNA used in Supplementary table 2.

As reported above to reviewer#1 BMP6 was upregulated in the f- progeny. Instead of upregulating the pathway with ligand gene expression, soluble BMP agonist was used in the antimyogenic cocktail. Correspondingly, a protein inhibitor of the pathway (Noggin) was used in the promyogenic cocktail as BMP6 was downregulated in the MAB- progeny. In combination with the RNAseq data, agonism and antagonism of the pathway with defined protein factors reliably implicates the BMP pathway in MiP regulation, although within the multi-factorial context of compound cocktails.

In Suppl 2 D and Fig 4G, the experimental protocol is complicated. Co-cultures were realized between C2C12 and the MiPs carrying a suicidal gene. Myogenic potentials of the MiPs were assessed by quantification of MyHC-positive area after exposure to AP20187 which induces death of MiP-C2C12 myotubes. There are 2 important biases in the analysis: 1) the test does not make the difference between a myotube exclusively composed of human nuclei and myotubes composed of both human and murine nuclei. 2) Differentiated mononuclear cells can be stained by the MF20 antibody.

It would have been more appropriate to co-culture the cells and performed the MF20 staining without AP20187 exposure. The number of myotubes containing 3 or more nuclei would have been counted as well as the number of

human nuclei per myotube. Human and murine nuclei can be very easily differentiated: C2C12 nuclei are characterized by the presence of heterochromatin revealed after Hoechst treatment or/and human nuclei are specifically recognized by the human specific LAMIN A/C antibody.

According to the reviewer 1 and 2 suggestions we performed again the analysis counting the number of myotubes (>3 nuclei) in each condition and the number of human nuclei per myotube. In this analysis we excluded MF20+ mononuclear cells as reported below and in Suppl Figure 1 and Figure 4 respectively in the revised ms.

MF20 labelling is not clear enough in Suppl 2. In Fig 4, in the presence of the vehicle, why is there no difference between f- and MAB-MiPs in the control wells? According to the previous article, human MAB-MiPs should show greater myogenic potential in co-culture with C2C12 than f-MiPs. This does not seem to be the case here and the way the graph is presented does not allow the reader to appreciate the intrinsic differences between the different points. In both Suppl 2 and Fig 4, in the MyHC+ graph, I would recommend to not normalize each control at 1 in order to be able to compare all the samples and to perform the adequate statistical test.

We thank the reviewer for this comment. We have now revised the ms with the new co-culture experiments reported above that allows a better understanding of the quantity of chimeric myotubes in all conditions and shows the increased myogenic potential of MAB-MiPs in co-culture with C2C12 vs f-MiPs. We have subsequently included the new co-culture analysis in Figure 4 and Suppl Figure 1 respectively, and we have created a new Supplementary Figure 4 with the experiments conducted in presence of the apoptotic drug

In the miRNA part, the authors have found 611 differentially expressed

miRNAs but again, the whole list is not provided. They have selected the miRNAs with predicted targets among the up-regulated genes previously identified per RNA-seq but the experiments performed to demonstrate the miRNAs really target these genes and are able to modulate their expression are not clear. Was it at mRNA level? Protein level? Both? Moreover, since one miRNA is able to modulate the expression of several genes, it would have been interesting to perform a RNA-seq on the treated cells to more precisely determine the genes that would be important for efficient muscle engraftment. A CpG/histone modification analysis after MiPs treatment with the miR and anti-miR cocktail might be interesting.

We included the list of differentially expressed miRNAs as Supplementary Table 4. We have crossed the miRNA sequencing data and the RNA sequencing data in order to putatively correlate genes and miRNA that were found differentially regulated. To this end we have checked. 3'UTR binding prediction and related mirsvr scores from microRNA.org. In Suppl Figure 3C we have validated our prediction at mRNA level via qPCR on treated and untreated cells.

According to the reviewer suggestions we performed RNA-seq on the treated cells to more precisely determine the genes that would be important for efficient muscle engraftment. We found several genes differentially expressed upon miRNA cocktail exposures. Among those, ANXA3, BMP6, MYB, LTBP4 were predicted target of miRNAs included in the cocktails. We would like to thank this reviewer that suggested this experiments that allowed us to identify additional genes modulated upon treatments and potentially involved in myogenic commitment of MiPS. In particular several genes coding for skeletal muscle and smooth muscle proteins were detected upregulated in PMC-treated samples or downregulated in AMC treated ones (ACTA2, ABLIM3, CNN2, CNN3).

We found that several elements of the BMP/TGF β pathways were upregulated upon AMC treatment. Among these we detected GDF15, that has been interestingly associated with muscle wasting in vivo (Patel et al JCSM, 2015- Garfield et al ERS, 2015).

We additionally identified several epigenetic regulators among differentially expressed genes after treatments such as, SMARCE1, HDAC6 and TET1. To this end we interestingly reported an upregulation of SAFB2 in pro-myogenic treated cells. It has been shown that its paralog SAFB1 facilitates the transition

of myogenic gene chromatin from a repressed to an activated state (Hernández-Hernández et al, Nuclei Acid res, 2013), our result raise the question of whether SAFB2 could be involved in similar processes. More details are provided in the results and in the discussion session (page 13 and page 17).

In addition, we analyzed the histone modifications and the CpG islands methylation after MiPs treatment with the miRNA and anti-miRNA cocktail, as reported below and in the new Figure 5 of the revised ms. Upon miRNA treatment the promoters of the 10 genes previously reported in Figure 3 again

showed some epigenetic signature in line with the myogenic propensity. Anti-myogenic gene pool BMP6, SMAD5 and MYB displayed a consistent pattern of DNA methylation and histone markers enrichment upon treatment suggesting a simultaneous multifactorial regulation of these promoters. ANXA3, LTBP4 and PAX7 did not show univocal signature at epigenetic level upon treatment, as the methylation status did not always correlate with the histone enrichment. This is in line with the general multifactorial mechanisms of action of epigenetic regulation, where several players contribute to maintain the transcriptionally active or inactive state of genes.

These results further explained and discussed are now included in the results and in the discussion section. (page 14 and 17)

We believe that this new set of data substantially improve the novelty of our work and better support our findings.

In the discussion, the authors should discuss the fact that the genes they have selected may be regulated by DNA methylation, histone modification and microRNA at the same time.

We thank the reviewer for this remark and suggestion and we have now edited the ms accordingly (page 17).

There are a huge numbers of errors in the manuscripts.

In the abstract: “we document that wildtype and genetically corrected MiPs can successfully engraft into the skeletal muscle and hearts of dystrophic dogs. ». There is nothing like that in the article. No dogs, no genetically corrected cells!!

We apologize for this mistake due to the previous version of the ms. We have

now removed this part an correct all the errors in the manuscripts.

The Supplementary figures are not well numbered. Moreover, the Suppl 2 is not mentioned in the text.

In figure 4, there is no point J. In Fig. 1, there is an unwanted picture right in the middle of the figure.

In figures, 1B, 1G and Suppl 1D, the text and the lines are not well aligned with the graphs.

We thank the reviewer for pointing out these errors and we have now proceeded to correct them in the revised ms.

In the material and methods, in the study approval section, “Injection of GRMD dogs and collection of canine samples was performed in compliance with the French law and the Ethical Approval of Maison Alford Veterinary School.” Again, there is no dog in this article.

Once again we apologize for the mistake and we have removed any references to dog from the current revised ms.

REVIEWERS' COMMENTS:

Reviewer #1 (Remarks to the Author):

The authors have replied to all my comments

--

Reviewer #2 (Remarks to the Author):

The article presented by Giacomazzi et al has been improved and most of the important points raised in the first review have been addressed.

However, concerning the CpG experiments, even if the most proximal CpG regions to the TSS were analyzed, this does not indicate that these CpGs are implicated in gene regulation (especially when only 8 CpGs are analyzed).

Concerning the miRNA part, there is still no experimental evidence that the miRNAs really target the up-regulated genes previously identified per RNA-seq. This should be specified.

In Fig1 D and E, please specify which skeletal muscles are analysed (hindlimbs).

In Fig 1F, GFP-positive cells are not easily visible.

We are grateful for the additional comment of the reviewers that contribute to improve the quality of our work. Answers to the referees are reported in *red italics*.

Itemized reply.

Reviewer #2 (Remarks to the Author):

The article presented by Giacomazzi et al has been improved and most of the important points raised in the first review have been addressed.

However, concerning the CpG experiments, even if the most proximal CpG regions to the TSS were analyzed, this does not indicate that these CpGs are implicated in gene regulation (especially when only 8 CpGs are analyzed).

We thank the reviewer for this remark, and we have now toned down the conclusions regarding CpG islands analysis, see line 395-398.

Concerning the miRNA part, there is still no experimental evidence that the miRNAs really target the up-regulated genes previously identified per RNA-seq. This should be specified.

We have included this observation, see line 236-237.

In Fig1 D and E, please specify which skeletal muscles are analysed (hindlimbs).

We have now specified the hindlimbs.

In Fig 1F, GFP-positive cells are not easily visible.

We have now added arrows to better highlight the GFP+ cells.